# REWARD ADAPTATION VIA Q-MANIPULATION

## ABSTRACT

In this paper, we propose a new solution to reward adaptation (RA), the problem where the learning agent adapts to a target reward function based on one or multiple existing behaviors learned a priori under the same domain dynamics but different reward functions. RA has many applications, such as adapting an autonomous driving agent that can already operate either fast (if transporting goods) or comfortable (if carrying passengers) to operating both fast and comfortable (if transporting goods with human passengers onboard). Learning the target behavior from scratch is possible but often inefficient given the available source behaviors. Our work represents a new approach to RA via the manipulation of Q-functions. Assuming that the target reward function is a known function of the source reward functions, our approach to RA computes bounds of the Q function. We introduce an iterative process to tighten the bounds, similar to value iteration. This enables action pruning in the target domain before learning even starts. We refer to such a method as "*Q-Manipulation*" (Q-M). We formally prove that our pruning strategy does not affect the optimality of the returned policy while empirically show that it improves the sample complexity. Comparison with baselines is performed in a variety of synthetic and simulation domains to demonstrate its effectiveness and generalizability.

## 1 INTRODUCTION

Reinforcement Learning (RL) as described by Watkins (1989); Sutton and Barto (2018) represents a class of learning methods that allow agents to learn from interacting with the environment. RL has demonstrated great successes in various domains such as games like Chess in Campbell et al. (2002), Go in Silver et al. (2016), and Atari games in Mnih et al. (2015), logistics in Yan et al. (2022), biology in Angermueller et al. (2019), and robotics in Kober et al. (2013). However, applying RL to many real-world problems still suffers from the issue of high sample complexity. Prior approaches have been proposed to alleviate the issue from different perspectives, such as learning optimization, transfer learning, modular and hierarchical RL, and offline RL.

The problem of reward adaptation (RA) was first introduced and addressed by Barreto et al. (2018; 2020), where the learning agent adapts to a target reward function given one or multiple existing behaviors learned a priori (referred to as the source behaviors) under the same actions and transition dynamics but different reward functions. RA has many useful applications, such as enabling a vehicle's driving behavior from two known behaviors (comfortable driving with passengers and fast driving for goods delivery) to a new target behavior that combines comfort and speed, accommodating both passengers and goods. Featuring such a special type of transfer learning, RA methods can benefit from an ever-growing repertoire of source behaviors to create new and potentially more complex target behaviors. Learning the target behavior from scratch is possible but often inefficient given the available source behaviors. In this paper, we present a new approach that offers its unique benefits compared to the previous work on RA.

To better conceptualize the RA problem, consider a grid-world as shown in Fig. 1, which is an expansion of the Dollar-Euro domain described by Russell and Zimdars (2003). In this domain, the agent can move to any of its adjacent locations at any step. The agent's initial location is colored in yellow and the terminal locations are colored pink or green, which correspond to the source reward functions (i.e., collecting dollars and euros), respectively. Visiting the terminal location with a single color returns a reward of $1.0$ under the corresponding reward function, and visiting the terminal location with split colors returns a reward of $0.6$ under both reward functions. In RA, the assumption

is that the optimal behaviors under the source reward functions are given, referred to as the source behaviors. A target domain may correspond to a reward function that awards both dollars and euros.

Under the assumption that the reward function is expressed in the form of feature weights such that the source behaviors can be evaluated easily under the target domain, prior work for addressing RA can be viewed as combining the best parts of the source behaviors to initialize learning, referred to as Successor Feature Q-Learning (SFQL) by Barreto et al. (2018; 2020). Consequently, SFQL may not work well for situations where the target behavior differs substantially from the source behaviors, such as in the Dollar-Euro domain. Our approach, instead, reasons about the best/worse-case scenarios under each source domain and combines such knowledge to compute upper/lower bounds of the target Q-function to enable action pruning. It results in a more general knowledge transfer method whose efficacy does not rely on the similarity between the source and target behaviors.

Our new approach to RA is referred to as "*Q-Manipulation*" (Q-M). We assume the existence of a function, referred to as the *combination function*, that relates the source reward functions to the target reward function. In practice, we often have a good idea about the functional relationship between the source and target reward functions (e.g., linear in the Dollar-Euro domain). Based on such a relationship, Q-M computes an upper and lower bound of Q-function in the target domain to identify actions that cannot contribute to the optimal behavior via an iterative process similar to value iteration. It enables us to prune those actions before learning the target behavior without affecting its optimality. In our evaluation, we empirically show that the effectiveness of

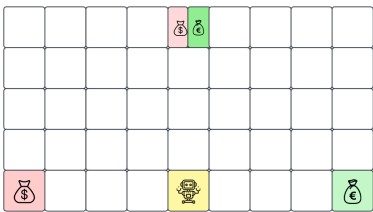

Figure 1: Dollar-Euro domain.

Q-M across simulated and randomly generated domains and analyze its limitations, focusing on conditions under which its efficacy is negatively impacted. Furthermore, we demonstrate that Q-M can still be effective in domains with continuous state spaces via discretization, even though the optimality guarantee would be lost there. In general, Q-M requires additional computing resources (i.e., CPU time and storage) to implement but its benefits outweigh the costs in practical applications, especially in situations where accessing the target domain for samples is expensive.

Our core contributions are: We address the problem of reward adaptation (RA) via Q-Manipulation (Q-M), which represents a new approach to RA that supports more general knowledge transfer than the previous work. In domains with discrete state spaces, we formally prove the correctness of the action pruning process under certain initialization conditions; otherwise, we suggest how Q-M may be applied to expedite learning at the cost of guaranteed optimality. We extensively evaluate Q-M with respect to baselines to validate its efficacy and analyze its limitations.

## 2 METHODOLOGY

In this section, we start with a brief introduction to reinforcement learning (RL) before discussing reward adaptation (RA) and our approach. In RL, the task environment is modeled as an MDP $M = (S, A, T, R, \gamma)$, where $S$ is the state space, $A$ is the action space, $T : S \times A \times S \rightarrow [0, 1]$ is the transition function, $R : S \times A \times S \rightarrow \mathbb{R}$ is the reward function, and $\gamma$ is the discount factor. At every step $t$, the RL agent observes state $s_t$ and takes an action $a_t \in A$. As a result, the agent progresses to state $s_{t+1}$ according to the transition dynamics $T(\cdot|s_t, a_t)$, and receives a reward $r_t$. The goal is to search for a policy that maximizes the expected cumulative reward or expected return. We use $\pi$ to denote a policy as a mapping from $S$ to $A$. The $Q$ function of the optimal policy $\pi^*$ is denoted by $Q^*$ and defined in Eq. 1. To prepare us for later discussion, we also introduce $Q^\mu$ (Eq. 2) to represent the $Q$ function of the "worst" policy that minimizes the expected return. The following lemma establishes the connection between $Q^\mu$ and a variant of $Q^*$:

$$Q^*(s, a) = \max_\pi \left[ \mathbb{E} \left[ \sum_{t=0}^{\infty} \gamma^t r_t | s_0, \pi \right] \right] \quad (1)$$

$$Q^\mu(s, a) = \min_\pi \left[ \mathbb{E} \left[ \sum_{t=0}^{\infty} \gamma^t r_t | s_0, \pi \right] \right] \quad (2)$$

**Lemma 1.**

$$Q_R^\mu(s, a) = -Q_{-R}^*(s, a) \quad (3)$$

*where $Q_{-R}^*(s, a)$ denotes the Q function of the optimal policy under negative R or $-R$.*

In this paper, we consider RL with discrete state and action spaces and deterministic policies. Extending the discussion to the continuous cases and stochastic policies will be future work. Proofs throughout the paper are included in the appendix.

## 2.1 REWARD ADAPTATION (RA)

**Definition 1** (Reward Adaptation (RA)). *Under $M \setminus R$, denoting an MDP without the specification of a reward function, RA is to determine the optimal policy for a target reward function $\mathcal{R}$, given a set of source behaviors trained under their respective source reward functions $R_1, R_2 \ldots R_n$.*

In RA, we assume the same transition dynamics, state and action spaces for the source and target behaviors. Note that the source domains are no longer accessible while learning the target behavior. Next, we provide the problem statement of RA under Q-M as follows:

**Problem Statement** [Reward Adaptation with $Q$-Variants]: *Given an RA problem where variants of the $Q$ functions are accessible for the source domains (e.g., $Q^*$'s and $Q^\mu$'s under the source reward functions), determine the optimal policy under a target reward function $\mathcal{R}$ that is a known function of the source reward functions specified as follows:*

$$\mathcal{R} = f(R_1, R_2, \ldots R_n) \tag{4}$$

$f$ above is also referred to as the combination function. When $f$ is not known exactly but can be modeled with an additional noise component, we will discuss later how Q-M can be adapted to handle such situations at the cost of reduced efficacy.

To derive a solution to RA with $Q$-variants, we propose Q-M, an action-pruning strategy that ensures that only unnecessary actions are pruned. To achieve this, we aim to compute an upper and lower bound of $Q^*$ under the target reward function based on the $Q$ variants from the source behaviors. Intuitively, if the lower bound of an action $a$ is higher than the upper bound of action $\hat{a}$ under a state $s$, $\hat{a}$ can be pruned. In Q-M, we derive these bounds based on an iterative process that we describe next.

## 2.2 Q-MANIPULATION

In Q-M, we first initialize an upper and lower bound of $Q^*_{\mathcal{R}}$ and then iteratively refine them. To avoid notation cluttering to improve clarity, we omit the subscript of $Q$ for indicating the reward function used. These two steps are formalized below (Note that we do not assume any knowledge of $Q^*$):

**Upper Bound (UB)**

$$Q_0^{UB}(s, a) > Q^* \qquad \text{[Initialization]} \tag{5}$$

$$Q_{k+1}^{UB}(s, a) = \min\left( Q_k^{UB}(s, a), \max_{s' \in \hat{T}(\cdot|s,a)} \left[ \mathcal{R}(s, a, s') + \gamma \max_{a'} Q_k^{UB}(s', a') \right] \right) \tag{6}$$

**Lower Bound (LB)**

$$Q_0^{LB} < Q^* \qquad \text{[Initialization]} \tag{7}$$

$$Q_{k+1}^{LB}(s, a) = \max\left( Q_k^{LB}(s, a), \min_{s' \in \hat{T}(\cdot|s,a)} \left[ \mathcal{R}(s, a, s') + \gamma \max_{a'} Q_k^{LB}(s', a') \right] \right) \tag{8}$$

$\hat{T}(\cdot|s, a)$ denotes **reachable states** (or neighbouring states) from $s, a$. This information is assumed to be available in Q-M or can be obtained while training the source behaviors. Similarly, the source reward functions or $R_i$'s are also assumed to be available so that $\mathcal{R}(s, a, s')$ in the equations above can be computed based on its known relationship with them (Eq. 4). The outermost max/min ensures $Q^{UB} \geq Q^* \geq Q^{LB}$ throughout the iterative processes via simple induction. It is worth noting that the updates above ensure that the upper and lower bounds are always decreasing and increasing, respectively, as desired such that the bounds are tightening. When the source reward functions are noisy, it requires their means to be used in the updates. Next, before discussing the initializations, we show that such processes converge to a fixed point in Q-M, respectively.

**Definition 2.** *The min and max Bellman operator for UB and LB in Q-M are mappings $\mathcal{T} : \mathbb{R}^{|S \times A|} \to \mathbb{R}^{|S \times A|}$ that satisfy, respectively:*

$$(\mathcal{T}_{min} Q_k^{UB})(s, a) = \min\left( Q_k^{UB}(s, a), \max_{s' \in \hat{T}(\cdot|s,a)} \left[ \mathcal{R}(s, a, s') + \gamma \max_{a'} Q_k^{UB}(s', a') \right] \right)$$

$$(\mathcal{T}_{max}Q_k^{LB})(s,a) = \max\left(Q_k^{LB}(s,a), \min_{s' \in \hat{T}(\cdot|s,a)}\left[\mathcal{R}(s,a,s') + \gamma \max_{a'} Q_k^{LB}(s',a')\right]\right)$$

Since the theoretical results for the min and max operator are similar, we do not distinguish between them below but provide separate proofs for them in the appendix.

**Theorem 1** (Convergence). *The iteration process introduced by the Bellman operator in Q-M satisfies*

$$\|\mathcal{T}Q_k - \mathcal{T}Q_{k+1}\|_\infty \le \gamma\|Q_k - Q_{k+1}\|_\infty, \forall Q_k, Q_{k+1} \in \mathbb{R}^{|S \times A|}.$$

*such that the Q function converges to a fixed point.*

Formally, $\|f\|_\infty = \sup_x |f(x)|$ and it returns the maximum absolute difference between $Q_k(s,a)$ and $Q_{k+1}(s,a)$ under any $s,a$ above. The process converges to a fixed point, since the difference between two consecutive iterations always decreases. However, it turns out that the fixed point may not necessarily be unique as with value iteration.

**Theorem 2.** *The Bellman operator in Q-M specifies only a non-strict contraction in general:*

$$\left\|\mathcal{T}Q - \mathcal{T}\hat{Q}\right\|_\infty \le \left\|Q - \hat{Q}\right\|_\infty$$

This result is interesting since it identifies another case where non-strict contraction results in a fixed point other than the identity map.

**Corollary 1** (Non-uniqueness). *The fixed point of the iteration process in Q-M may not be unique.*

In our evaluation, we observe that the fixed point found by the iteration process depends on the initialization. Another observation is that the Bellman operator in Q-M appears almost identical to that in value iteration when the MDP is deterministic. In such cases, we observe that Q-M often results in zero-shot learning when the upper and lower bounds converge to $Q_\mathcal{R}^*$.

## 2.3 INITIALIZING THE BOUNDS

A simple way to initialize the bounds would be to identify the most positive and negative rewards and compute the sums of their geometric sequences via the discount factor, respectively. However, these bounds are likely to be too conservative to be useful since the iteration processes may converge undesirably due to non-unique fixed points. Intuitively, we would like the bounds to be tight initially to yield the best results. However, computing bounds for the target behavior based on information from the source behaviors only is not a trivial task. Next, we show situations where additional assumptions hold such that we can provide more desirable initializations. In particular, we will show next how different forms of the combination function $f$ in Eq. 4 can affect the initializations.

**Linear Combination Function:** First, we consider the case when the target reward function is a linear function of the source reward functions. In such cases, if the agent maintains both $Q_i^\mu$'s and $Q_i^*$'s while learning the source behaviors, we propose the initializations as follows. Note that $Q_i^\mu$ can be obtained conveniently while learning the source behaviors based on Lemma 1.

**Lemma 2.** *When $\mathcal{R} = \sum c_i R_i$ where $c_i \ge 0$ , an upper and lower bound of $Q_\mathcal{R}^*$ are given, respectively, by:*

$$Q_0^{UB} = \sum_{i=1}^n c_i Q_i^*$$
$$Q_0^{LB} = \max_i[c_i Q_i^* + \sum_j c_j Q_j^\mu] \ where \ j \in \{1:n\} \setminus i \tag{9}$$

**Nonlinear Combination Function:** Handling nonlinear combination is more complicated and deriving tight bounds that are guaranteed to be correct is difficult. Instead, we propose approximate bounds for monotonically increasing and positive function $f$ as follows:

$$Q_0^{UB} = f(Q_{|R_1|}^*, Q_{|R_2|}^*, \dots Q_{|R_n|}^*) \qquad Q_0^{LB} = -f(Q_{|R_1|}^*, Q_{|R_2|}^*, \dots Q_{|R_n|}^*) \tag{10}$$

Using the bounds above requires the agent to maintain $Q_{|R_i|}^*$'s. Since these bounds are approximate, they do not guarantee correctness in general, meaning that actions belonging to the optimal policy may be pruned. However, we show that they work well in practice in our evaluation.

## 2.4 Noisy Combination Function and Continuous State Spaces

**Noisy Combination Function:** When the combination function is not known exactly but can be modeled with an additional noise component such that $\mathcal{R} = f(R_1 \ldots R_n) + N$, and we know the range of the noise (i.e., $N_{min}$ and $N_{max}$). We can consider such situations by augmenting the $\mathcal{R}(s, a, s')$ in Eqs. 6 and 8 with $N_{max}$ and $N_{min}$, respectively. We must also update the initialization of the bounds using $Q^{UB} = Q^{UB} + N_{max} \times \frac{1-\gamma^{t_{max}}}{1-\gamma}$ and $Q^{LB} = Q^{LB} + N_{min} \times \frac{1-\gamma^{t_{max}}}{1-\gamma}$, where $t_{max}$ is the maximum steps in an episode. Note however that such modifications will likely reduce the efficacy of Q-M.

**Handling Continuous State Spaces:** For domains with continuous state spaces, we resort to using features (e.g., tile-coding) to discretize the state space and then apply the process of Q-M on such a space to prune actions. We can then run any RL method that can handle continuous state spaces (such as Deep Q-Learning) under the reduced action space per each discrete state. Although the optimality guarantee is obviously lost due to the discretization, we aim to show how effective such a simple adaption can be. The implementation details are discussed in Sec. 3. We will extend Q-M to natively handle continuous state and action spaces in future work.

**Action Pruning in Q-M:** Intuitively, if an action $a$'s lower bound is higher than some other action $\hat{a}$'s upper bound under a state $s$, then $\hat{a}$ can be pruned for that state. This allows us to reduce the action space per each different state, which contributes to faster convergence. When the upper and lower bounds are sound, the optimal policies are preserved.

**Theorem 3.** *[Optimality] For reward adaptation with Q variants, the optimal policies in the target domain remain invariant under Q-M when the upper and lower bounds are initialized correctly.*

## 3 Evaluation

The primary objective here is to evaluate the performance of Q-M to analyze its benefits and limitations. We compare Q-M with SFQL described by Barreto et al. (2018), the state-of-the-art approach to reward adaptation. Q-M and SFQL initialize learning in different ways to transfer prior knowledge from the source domains but otherwise both implement Q-Learning (QL) to learn the target behavior. Hence, we also use QL without any knowledge transfer as a baseline. More specifically, to initialize learning for SFQL, we evaluate the given source behaviors on the target domain to compute a bootstrap $Q$-function as described in the generalized policy improvement theorem in Barreto et al. (2018). Additional results analyzing Q-M (including where actions are pruned) and running time comparisons are reported in Sec. A.3. We keep the hyperparameters for Q-Learning (or DQN) the same across the different methods.

Since we are interested in demonstrating Q-M as a more general knowledge transfer method than SFQL, we design the evaluation domains such that the target behaviors are substantially different from the source behaviors in most of them (similar to the situation in Dollar-Euro). In such cases, SFQL, initializing learning by combining the best parts of the source behaviors, is expected to not perform well unless the target behavior happens to be characterized by some combination of the source behaviors. Details on how the source and target behaviors are designed are in the appendix.

For Q-M, we use the initializations described in Sec. 2.3. One observation about Q-M is that the computation of UB and LB is affected substantially by the stochastic branching factor (SBF) of a domain, as evident in Eqs. 6 and 8. SBF here is defined as the maximum number of next states reachable (or with a nonzero transition probability) from any state and action pair. Intuitively, the less stochastic the domain is, the more the Bellman updates in Q-M resemble that in value iteration (except for the outermost max/min). To demonstrate the influence of SBF, for each evaluation domain, we gradually increase its SBF. At the same time, the number of reachable states from a given state and action pair is allowed to vary and randomly chosen between 1 and a set SBF. We first evaluate with simulation and randomly generated domains under linear combination functions and then move on to the more challenging cases of nonlinear and noisy functions. To showcase the generality of Q-M, we also consider randomizing the domains so that we evaluate with 1) given MDP $\setminus R$ and designed rewards, 2) randomized MDP $\setminus R$ and designed rewards, and 3) randomized MDP $\setminus R$ and randomized rewards. All evaluations are averaged over 30 runs. More details about the evaluation settings along with a detailed description of all the domains, including the design of source and target behaviors, are reported in the appendix.

## 3.1 LINEAR COMBINATION FUNCTION

**Given MDP $\setminus R$ and Designed Rewards:**

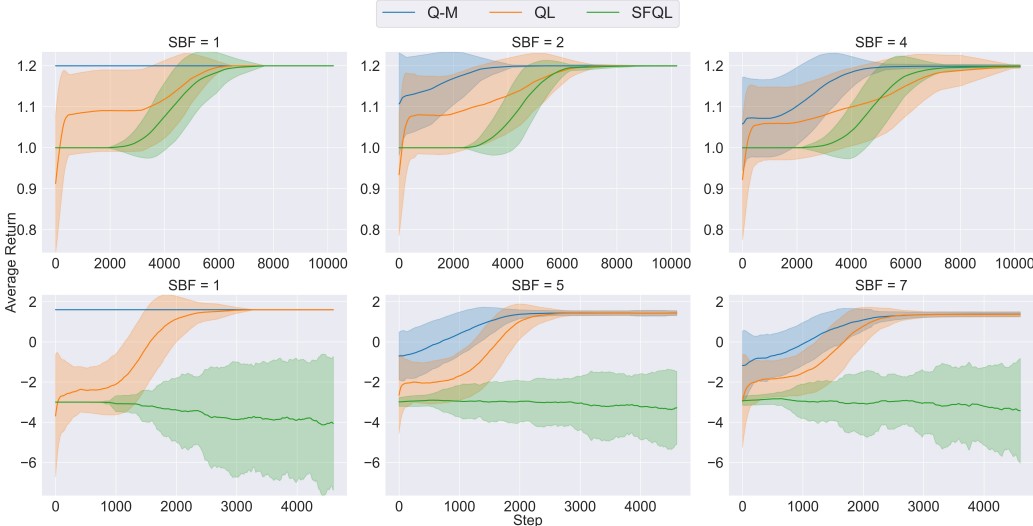

Figure 2: Convergence plots for Dollar Euro (top) and Racetrack (bottom).

In this evaluation, we compared Q-M with the baselines in simulation domains that include Racetrack and Dollar-Euro. The convergence plots are shown in Fig. 2. In each subfigure, we show the SBF used (labeled at the top). We observe that Q-M converges substantially faster than the baselines in both domains. However, as expected, the performance of Q-M is negatively impacted as SBF increases. An interesting observation is the performance of SFQL. SFQL seems to struggle with these domains, especially Racetrack. Since the sources behaviors differ much from the target behavior, knowledge transfer in SFQL based on combining the source behaviors can actually misguide the learning process. It is worth mentioning that SFQL eventually converged to the optimal policy after we allowed it to train with more episodes. In addition, we also observe that Q-M in deterministic scenarios (left most subfigures when SBF = 1) results in zero-shot learning: its iterative processes for computing UB and LB both converge to $Q_{\mathcal{R}}^*$. This result demonstrates that Q-M is indeed a more general knowledge transfer method that does not depend on the similarity between the source and target behaviors.

**Randomized MDP $\setminus R$ and Designed Rewards:** First, we evaluated with the Frozen Lake domain

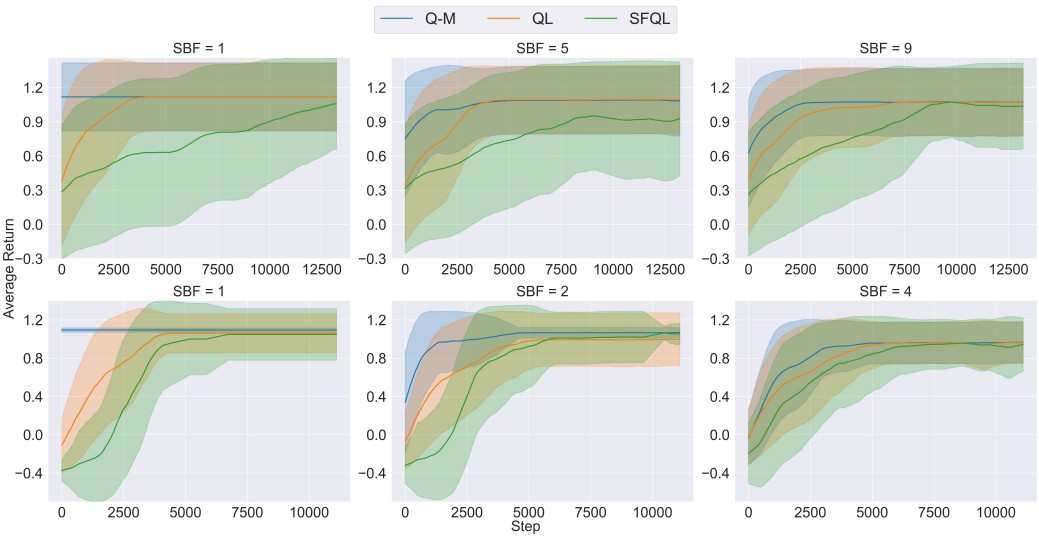

Figure 3: Convergence plots for auto-generated domains (top) and Frozen Lake (bottom).

while randomizing the hole locations (4 holes) in each run. Additionally, we evaluated with auto-generated MDP$\backslash R$'s where the numbers of states and actions are randomly generated, and terminal states were randomly selected. The number of terminal states in both domains was held fixed as well as their terminal rewards. The convergence plots are presented in Fig. 3. Similarly, we can observe that Q-M performs the best in both domains. It demonstrates that Q-M can generalize to different configurations of MDP$\backslash R$.

**Randomized MDP $\backslash R$ and Randomized Rewards:** In this evaluation, we aim to push the results

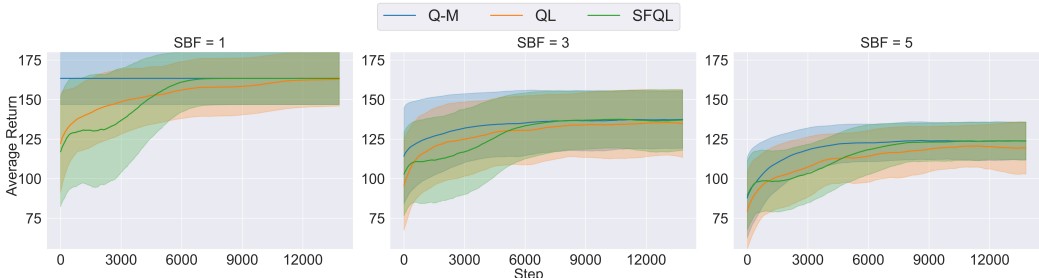

Figure 4: Convergence plots for auto-generated domains.

from the previous evaluation further by analyzing the generality of Q-M with both randomized MDP $\backslash R$ and rewards. Randomizing all of these factors simultaneously can introduce very different behaviors, which represent more challenging situations to generalize. In this evaluation, MDP$\backslash R$'s with fixed numbers of states and actions were auto-generated in each run. A fixed number of terminal states were selected randomly. Rewards for each transition, including terminal states, were generated randomly. The convergence plots are presented in Fig. 4. Q-M still consistently performs better than the baselines. However, we can also observe that SFQL performs better than QL, which is in contrast to the previous evaluations. This is likely due to the fact that a high level randomization here results in more similarities between the source and target behaviors that are taken advantage of by SFQL.

### 3.2 NONLINEAR COMBINATION FUNCTION

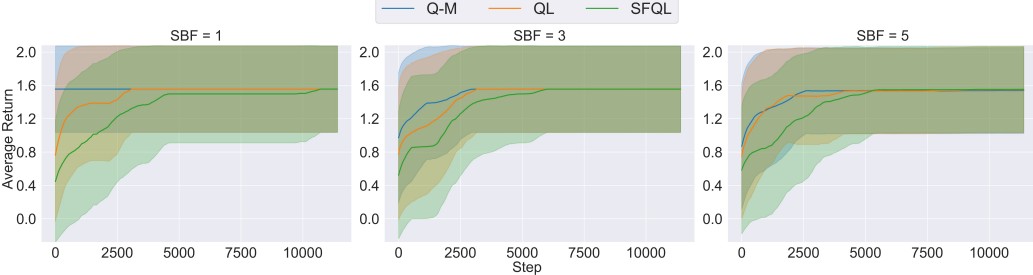

Figure 5: Convergence plots for auto-generated domains with a nonlinear $f$: $\mathcal{R} = (R_1 + R_2)^3$.

We now extend our evaluation to nonlinear combination functions. The main aim here is to evaluate the effectiveness of the initializations proposed even though the optimality guarantee is lost. In this evaluation, we use the same setting as in Randomized MDP$\backslash R$ and Designed Rewards above. The convergence plots are presented in Fig. 5. We observe that Q-M is still more efficient than the baselines although the performance gain is not as obvious as in the previous evaluations, especially as shown in the last subfigure. As expected, RA with nonlinear combination functions is to more challenging than with linear functions, resulting in reduced action pruning. This is due in part to the difficulty in establishing bounds that are tight while being sound.

### 3.3 NOISY COMBINATION FUNCTION

We aim to evaluate how Q-M would perform under noisy combination functions and how noise affects its performance. We used the same setting as in Randomized MDP$\backslash R$ and Randomized Rewards above. We consider a situation where the combination function is not exactly known but can be modeled by using a noise component: $\mathcal{R} = R_1 + R_2 + N$. Assuming the knowledge of $N_{\min}$

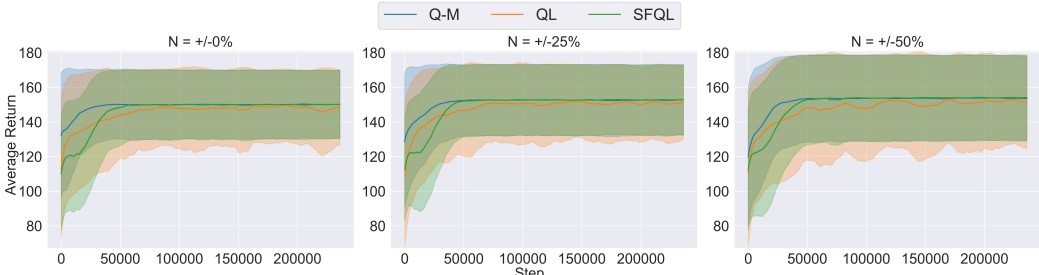

Figure 6: Convergence plots with auto-generated domains and a noisy combination function.

and $N_{\max}$, we updated the initializations and Bellman updates for Q-M. The convergence plots are presented in Fig. 6 where the noise levels with respect to the mean of the rewards was labeled at the top. As expected, we observe that noise has an impact on the efficacy of Q-M: the more noise, the smaller the performance gain with respect to the baselines. However, it is promising to observe that Q-M can still be effective under such noisy situations since it can greatly expand the applicability of Q-M. For instance, when the functional relationship is unknown, we can apply regression to fit the source reward functions to the observed target rewards under an assumed functional form based on domain expertise; noise can be incorporated to handle regression error.

## 3.4 DOMAINS WITH CONTINUOUS STATE SPACES

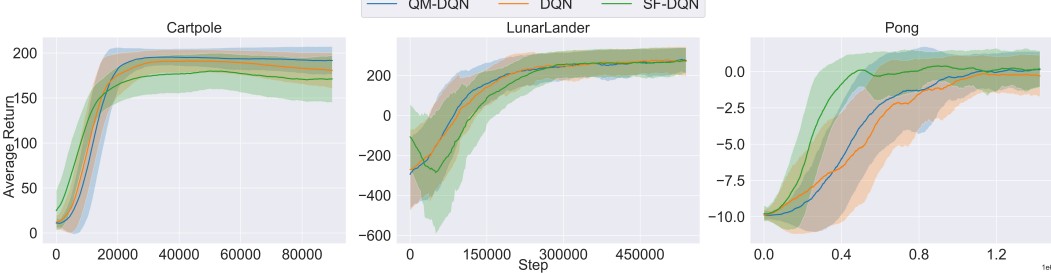

Figure 7: Convergence plots for domains with continuous state spaces.

In environments with continuous state spaces, we applied both Q-M and SFQL with discretized state spaces based on tile-coding, where each feature is discretized to produce the state space. The source $Q$-functions are also discretized with values determined according to the midpoint of each discrete state. For Q-M, we also maintained a fixed number of reachable states from any state and action pair (assumed to be given or learned from training source behaviors) to compute the Bellman updates. We used Deep Q-Network (DQN) as the underlying learning method after initializing learning for both Q-M and SFQL. During learning in Q-M, pruned actions in a discrete state are not considered for any state belonging to that state. Convergence plots are presented in Fig. 7. We observe that Q-M (QM-DQN) performs only marginally better than the baselines in Cartpole and Lunar Lander, suggesting that discretization has a significant negative impact on the performance of Q-M. This is expected since discretization has the effect of adding substantial "noise" to the $Q$ functions. It is however encouraging to see that Q-M in such cases seems to have avoided pruning out the optimal actions. In Pong, SF-DQN outperformed both QM-DQN and DQN. This was due to the choice of source behaviors that are either keeping left or right. The target behavior requires the agent to move to the left and right to catch the ball, which shares strong similarity with the source behaviors.

## 4 RELATED WORK

**Reward and Q-Decomposition**: The combination function in Q-M can be viewed in general as specifying a structure of the target reward function based on the source functions. Reward structure can significantly influence the effectiveness of an RL agent as discussed in Silver et al. (2021). Prior approaches such as Lin et al. (2019); Marthi (2007); Ciardo and Trivedi (1993) have suggested novel ways to exploit reward structure and decompose the reward function to better learn. For example, Q-Decomposition as described by Russell and Zimdars (2003) involves a similar setting to ours

where it aims to learn a behavior under a reward function that is the linear sum of multiple sub-reward functions. Each sub-agent for such a sub-reward function undergoes its own learning process and supplies its Q values to an aggregator. The idea has also been extended to work with Deep Q Networks (DQN) by Van Seijen et al. (2017). There, it is argued that reward decomposition enables faster learning as separate value functions only depend on a subset of input features, resulting in simpler domains. Similar ideas are developed in Sutton et al. (2011); Sprague and Ballard (2003). While these ideas are inspirational to ours, they are mostly for learning from scratch. No transfer is considered.

**Multi-Objective Reinforcement Learning**: Multi-Objective Reinforcement Learning (MORL) as described in Liu et al. (2014); Sprague and Ballard (2003); Roijers et al. (2013); Vamplew et al. (2011) is a branch of RL that deals with learning trade-offs between multiple objectives. A common approach to MORL is to search for the Pareto frontier, which is generally infeasible. A more practical way to combine the objectives uses linear scalarization as discussed by Van Moffaert et al. (2013). Often, the domain expert decides the weights for the objectives. Limitations have been reported by Vamplew et al. (2008) and solutions to counter them are proposed such as using the Chebyshev function. Our problem setting can be considered as a special case of MORL where the different objectives must be combined in complex ways. However, our focus is on improving sample complexity during learning by utilizing the existing behaviors for the individual objectives.

**Hierarchical Reinforcement Learning**: Hierarchical RL (HRL) as discussed in Dietterich (1998); Vezhnevets et al. (2017); Barreto et al. (2020); Bacon et al. (2017); Barto and Mahadevan (2003); Xiaoqin et al. (2009); Cai et al. (2013); Doroodgar and Nejat (2010) is the process of learning based on a hierarchy of behaviors that is often assumed to be known or learned. A hierarchical structure makes it possible to divide a learning problem into sub-problems, sometimes in a recursive manner. At any point in time, a hierarchy of behaviors may be activated and the behavior at the lowest level determines the output behavior. In HRL, the interaction between the behaviors is often assumed to be simple, i.e., sequential execution, since they are considered to address different parts of the state space. In contrast, the source and target behaviors in our work share the same state and action spaces and their interactions can be arbitrarily complex via the correlations between their reward functions.

**Transfer Learning and Multi-Task Learning**: Transfer learning, with various applications such as those described in Andreas et al. (2016); Bahdanau et al. (2016); Chang et al. (2015), is the process of learning a target task by leveraging experiences from source tasks. As a transfer learning method for reinforcement learning, multi-task reinforcement learning surveyed in Vithayathil Varghese and Mahmoud (2020) deals with learning from multiple related tasks simultaneously to expedite learning. In D'Eramo et al. (2019), for instance, individual learning agents learn from a related task and share their weights with the global network at regular intervals. The global network also periodically shares its parameters with individual learning agents. Our approach also deals with knowledge transfer from the source to the target domains. However, it represents the class of indirect transfer methods where the agent must "infer" useful information from the given information (i.e., source behaviors) before using it. Furthermore, in contrast to domain adaptation discussed in Peng et al. (2018); Eysenbach et al. (2020) for addressing the sim-to-real gap, reward adaptation is more about transferring knowledge between different tasks (i.e., reward functions).

## 5 CONCLUSIONS

In this paper, we introduced reward adaptation, the problem where the learning agent adapted to a target reward function based on the existing source behaviors under the same MDP $\setminus R$. We proposed an approach to reward adaptation, referred as Q-Manipulation (Q-M). The key was to maintain $Q$ variants for each of the source behaviors and apply Q-M iterations to compute bounds of the target $Q$ function and their initializations for action pruning before learning the target behavior. We formally proved that our approach converged and retained optimality under correct initializations. Empirically, we showed that Q-M was substantially more efficient than the baselines in domains where the source and target behaviors differ, and generalizable under different randomizations. We also applied Q-M to noisy combination functions and continuous state spaces to extend its applicability. As such, Q-M represents a valuable contribution to advancing transfer learning for reinforcement learning. It is worth mentioning that, given its unique way of knowledge transfer, Q-M can be combined with other approaches (such as SFQL) to further improve learning. Our work also opens up many future research opportunities, such as addressing continuous state and action spaces and handling different domain dynamics (in addition to difference in reward functions) as in domain adaptation.

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

## A APPENDIX

### A.1 THEORETICAL PROOFS

**Lemma 1**

$$Q_R^\mu(s,a) = \min_\pi \left[ \mathbb{E}\left[ \sum_{t=0}^\infty \gamma^t r_t | s_0, \pi \right] \right]$$
$$= -\max_\pi \left[ \mathbb{E}\left[ \sum_{t=0}^\infty -\gamma^t r_t | s_0, \pi \right] \right] \tag{11}$$
$$= -Q_{-R}^*(s,a)$$

**Lemma 2** When $\mathcal{R} = \sum c_i R_i$ where $c_i \geq 0$, an upper and lower bound of $Q_\mathcal{R}^*$ are given, respectively, by:

$$Q_0^{UB} = \sum_{i=1}^n c_i Q_i^*$$
$$Q_0^{LB} = \max_i [c_i Q_i^* + \sum_j c_j Q_j^\mu] \quad \text{where } j \in \{1:n\} \setminus i \tag{12}$$

*Proof.* From definition, we have:

$$c_i Q_i^\pi = \max_\pi \left[ \mathbb{E}\left[ c_i r_{i,0} + \gamma c_i r_{i,1} + \ldots + \gamma^n c_i r_{i,n} | s_0, \pi \right] \right] \tag{13}$$

By reorganizing the reward components, we have:

$$\sum_i c_i Q_i^\pi = Q_{\sum_i c_i R_i}^\pi \tag{14}$$

Denote the optimal policy under the target reward function $\mathcal{R}$ as $\pi^*$, given $c_i \geq 0$, we can derive that

$$\sum_i c_i Q_i^* \geq \sum_i c_i Q_i^{\pi^*} = Q_\mathcal{R}^* \tag{15}$$

For the lower bound, we have:

$$\max_i (c_i Q_i^* + \sum_{j \neq i} c_j Q_j^\mu) \leq c_k Q_k^* + \sum_{j \neq k} c_j Q_j^{\pi_k^*}$$

where $k$ denotes the best choice of $i$ from the left

$$\leq \max_\pi (c_i Q_i^\pi + \sum_{j \neq i} c_j Q_j^\pi) \tag{16}$$

$$= Q_\mathcal{R}^*$$

$$\square$$

Next, we present a few lemmas that are used in the proof of our theorems:

**Lemma 3.**

$$\left| \max_a f(a) - \max_a g(a) \right| \leq \max_a |f(a) - g(a)|.$$

*Proof.* Assume without loss of generality that $\max_a f(a) \geq \max_a g(a)$, and denote $a^* = \arg\max_a f(a)$. Then,

$$\left|\max_a f(a) - \max_a g(a)\right| = \max_a f(a) - \max_a g(a) = f(a^*) - \max_a g(a) \leq f(a^*) - g(a^*) \leq \max_a |f(a) - g(a)|.$$

This concludes the proof. □

**Lemma 4.**
$$\left|\min_a f(a) - \min_a g(a)\right| \leq \max_a |f(a) - g(a)|.$$

*Proof.* Assume without loss of generality that $f(a^*) = \min_a f(a) \geq \min_a g(a) = g(b^*)$. Then,

$$\max_a |f(a) - g(a)| \geq |f(b^*) - g(b^*)| \geq f(b^*) - g(b^*) \geq f(a^*) - g(b^*) = \left|\min_a f(a) - \min_a g(a)\right|$$

This concludes the proof. □

**Theorem 1** [Convergence] The iteration process introduced by the Bellman operator in Q-M satisfies

$$\|\mathcal{T}Q_k - \mathcal{T}Q_{k+1}\|_\infty \leq \gamma\|Q_k - Q_{k+1}\|_\infty, \forall Q_k, Q_{k+1} \in \mathbb{R}^{|S \times A|}.$$

such that the $Q$ function converges to a fixed point.

*Proof.* **1) Upper Bound**

The operator $\mathcal{T}_{\min}$ for the upper bound is defined as follows:

$$Q_{k+1}^{UB}(s,a) = (\mathcal{T}_{\min}Q_k^{UB})(s,a) = \min\left(Q_k^{UB}(s,a), \max_{s' \in \hat{T}(\cdot|s,a)}\left[\mathcal{R}(s,a,s') + \gamma \max_{a'} Q_k^{UB}(s',a')\right]\right)$$
(17)

where $\hat{T}(\cdot|s,a)$ denotes reachable states from $s, a$.

We consider the change of difference between $Q$ values between before and after the modified Bellman update (i.e., the difference between $\left|Q_k^{UB}(s,a) - Q_{k+1}^{UB}(s,a)\right|$ and $\left|Q_{k+1}^{UB}(s,a) - Q_{k+2}^{UB}(s,a)\right|$):

**Case 1:** If the first elements were the smaller values for computing both $Q_{k+1}^{UB}$ and $Q_{k+2}^{LB}$ in Eq. 17:

$$Q_{k+1}^{UB}(s,a) = Q_k^{UB}(s,a)$$

$$Q_{k+2}^{UB}(s,a) = Q_{k+1}^{UB}(s,a)$$

$$\left|Q_{k+1}^{UB}(s,a) - Q_{k+2}^{UB}(s,a)\right| = |Q_k^{UB}(s,a) - Q_{k+1}^{UB}(s,a)| = 0$$

**Case 2:** If the second element in min was the smaller value for computing $Q_{k+1}^{UB}$ and the first element in min was the smaller value for $Q_{k+2}^{UB}$:

$$Q_{k+1}^{UB}(s,a) = \max_{s' \in \hat{T}(\cdot|s,a)}\left[\mathcal{R}(s,a,s') + \gamma \max_{a'} Q_k^{UB}(s',a')\right]$$

$$Q_{k+2}^{UB}(s,a) = Q_{k+1}^{UB}(s,a)$$

$$\left|Q_{k+1}^{UB}(s,a) - Q_{k+2}^{UB}(s,a)\right| = 0$$

**Case 3:** If the first element in min was the smaller value for computing $Q_{k+1}^{UB}$ and the second element in min was the smaller value for $Q_{k+2}^{UB}$:

$$Q_{k+1}^{UB}(s,a) = Q_k^{UB}(s,a) \leq \max_{s' \in \hat{T}(\cdot|s,a)}\left[\mathcal{R}(s,a,s') + \gamma \max_{a'} Q_k^{UB}(s',a')\right] \text{ (Eq. 17)} \quad (18)$$

$$Q_{k+2}^{UB}(s,a) = \max_{s' \in \hat{T}(\cdot|s,a)}\left[\mathcal{R}(s,a,s') + \gamma \max_{a'} Q_{k+1}^{UB}(s',a')\right]$$

$$\left|Q_{k+1}^{UB}(s,a) - Q_{k+2}^{UB}(s,a)\right|$$

$$= Q_k^{UB}(s,a) - \max_{s'\in\hat{T}(\cdot|s,a)}\left[\mathcal{R}(s,a,s') + \gamma\max_{a'}Q_{k+1}^{UB}(s',a')\right]$$

$$\leq \max_{s'\in\hat{T}(\cdot|s,a)}\left[\mathcal{R}(s,a,s') + \gamma\max_{a'}Q_k^{UB}(s',a')\right] - \max_{s'\in\hat{T}(\cdot|s,a)}\left[\mathcal{R}(s,a,s') + \gamma\max_{a'}Q_{k+1}^{UB}(s',a')\right] \text{ (Eq. 18)}$$

$$\leq \left|\max_{s'\in\hat{T}(\cdot|s,a)}\left[\mathcal{R}(s,a,s') + \gamma\max_{a'}Q_k^{UB}(s',a')\right] - \max_{s'\in\hat{T}(\cdot|s,a)}\left[\mathcal{R}(s,a,s') + \gamma\max_{a'}Q_{k+1}^{UB}(s',a')\right]\right|$$

$$\leq \gamma\max_{s'\in\hat{T}(\cdot|s,a)}\left|\max_{a'}Q_k^{UB}(s',a') - \max_{a'}Q_{k+1}^{UB}(s',a')\right| \text{ (Lemma 3)}$$

$$\leq \gamma\max_{s'\in\hat{T}(\cdot|s,a)}\max_{a'}\left|Q_k^{UB}(s',a') - Q_{k+1}^{UB}(s',a')\right| \text{ (Lemma 3)}$$

$$\leq \gamma\|Q_k^{UB}(s,a) - Q_{k+1}^{UB}(s,a)\|_\infty$$

**Case 4:** If the second elements in min were the smaller values for both $Q_{k+1}^{UB}$ and $Q_{k+2}^{UB}$:

$$Q_{k+1}^{UB}(s,a) = \max_{s'\in\hat{T}(\cdot|s,a)}\left[\mathcal{R}(s,a,s') + \gamma\max_{a'}Q_k^{UB}(s',a')\right]$$

$$Q_{k+2}^{UB}(s,a) = \max_{s'\in\hat{T}(\cdot|s,a)}\left[\mathcal{R}(s,a,s') + \gamma\max_{a'}Q_{k+1}^{UB}(s',a')\right]$$

$$\left|Q_{k+1}^{UB}(s,a) - Q_{k+2}^{UB}(s,a)\right|$$

$$= \left|\max_{s'\in\hat{T}(\cdot|s,a)}\left[\mathcal{R}(s,a,s') + \gamma\max_{a'}Q_k^{UB}(s',a')\right] - \max_{s'\in\hat{T}(\cdot|s,a)}\left[\mathcal{R}(s,a,s') + \gamma\max_{a'}Q_{k+1}^{UB}(s',a')\right]\right|$$

$$\leq \gamma\|Q_k^{UB}(s,a) - Q_{k+1}^{UB}(s,a)\|_\infty \text{ (similar to Case 3 above)}$$

Since the above cases hold for any $s,a$, we therefore have:

$$\|Q_{k+1}^{UB} - Q_{k+2}^{UB}\|_\infty \leq \gamma\|Q_k^{UB} - Q_{k+1}^{UB}\|_\infty \tag{19}$$

Since the distance decreases by gamma with every iteration, it will converge to $0$ and hence $Q^{UB}$ converges to a fixed point.

**2) Lower Bound**

The operator $\mathcal{T}_{\max}$ for the lower bound is defined as follows:

$$Q_{k+1}^{LB}(s,a) = (\mathcal{T}_{\max}Q_k^{LB})(s,a) = \max\left(Q_k^{LB}(s,a), \min_{s'\in\hat{T}(\cdot|s,a)}\left[\mathcal{R}(s,a,s') + \gamma\max_{a'}Q_k^{LB}(s',a')\right]\right)$$
$$\tag{20}$$

$\hat{T}(\cdot|s,a)$ denotes reachable states from $s,a$.

We consider the change of difference between Q values between before and after the modified Bellman update (i.e., the difference between $\left|Q_k^{LB}(s,a) - Q_{k+1}^{LB}(s,a)\right|$ and $\left|Q_{k+1}^{LB}(s,a) - Q_{k+2}^{LB}(s,a)\right|$):

**Case 1:** If the first elements in max were the bigger values for both $Q_{k+1}^{LB}$ and $Q_{k+2}^{LB}$:

$$Q_{k+1}^{LB}(s,a) = Q_k^{LB}(s,a)$$

$$Q_{k+2}^{LB}(s,a) = Q_{k+1}^{LB}(s,a)$$

$$\left|Q_{k+1}^{LB}(s,a) - Q_{k+2}^{LB}(s,a)\right| = |Q_k^{LB}(s,a) - Q_{k+1}^{LB}(s,a)| = 0$$

**Case 2:** If the second element in max was the bigger value for $Q_{k+1}^{LB}$ and the first element in max was the bigger value for $Q_{k+2}^{LB}$:

$$Q_{k+1}^{LB}(s,a) = \min_{s' \in \hat{T}(\cdot|s,a)} \left[ \mathcal{R}(s,a,s') + \gamma \max_{a'} Q_k^{LB}(s',a') \right]$$

$$Q_{k+2}^{LB}(s,a) = Q_{k+1}^{LB}(s,a)$$

$$\left| Q_{k+1}^{LB}(s,a) - Q_{k+2}^{LB}(s,a) \right| = 0$$

**Case 3:** If the first element in max was the bigger value for $Q_{k+1}^{LB}$ and the second element in max was the bigger value for $Q_{k+2}^{LB}$:

$$Q_{k+1}^{LB}(s,a) = Q_k^{LB}(s,a) \geq \min_{s' \in \hat{T}(\cdot|s,a)} \left[ \mathcal{R}(s,a,s') + \gamma \max_{a'} Q_k^{LB}(s',a') \right] \tag{21}$$

$$Q_{k+2}^{LB}(s,a) = \min_{s' \in \hat{T}(\cdot|s,a)} \left[ \mathcal{R}(s,a,s') + \gamma \max_{a'} Q_{k+1}^{LB}(s',a') \right]$$

$$\left| Q_{k+1}^{LB}(s,a) - Q_{k+2}^{LB}(s,a) \right|$$

$$= - \left( Q_k^{LB}(s,a) - \min_{s' \in \hat{T}(\cdot|s,a)} \left[ \mathcal{R}(s,a,s') + \gamma \max_{a'} Q_{k+1}^{LB}(s',a') \right] \right)$$

$$\left( \text{since } Q_{k+2}^{LB}(s,a) \geq Q_{k+1}^{LB}(s,a) \text{ based on Eq. 20} \right)$$

$$\leq - \left( \min_{s' \in \hat{T}(\cdot|s,a)} \left[ \mathcal{R}(s,a,s') + \gamma \max_{a'} Q_k^{LB}(s',a') \right] - \min_{s' \in \hat{T}(\cdot|s,a)} \left[ \mathcal{R}(s,a,s') + \gamma \max_{a'} Q_{k+1}^{LB}(s',a') \right] \right) \text{ (Eq. 21)}$$

$$\leq \left| \min_{s' \in \hat{T}(\cdot|s,a)} \left[ \mathcal{R}(s,a,s') + \gamma \max_{a'} Q_k^{LB}(s',a') \right] - \min_{s' \in \hat{T}(\cdot|s,a)} \left[ \mathcal{R}(s,a,s') + \gamma \max_{a'} Q_{k+1}^{LB}(s',a') \right] \right|$$

$$\leq \gamma \max_{s' \in \hat{T}(\cdot|s,a)} \left| \max_{a'} Q_k^{LB}(s',a') - \max_{a'} Q_{k+1}^{LB}(s',a') \right| \text{ (Lemma 4)}$$

$$\leq \gamma \max_{s' \in \hat{T}(\cdot|s,a)} \max_{a'} \left| Q_k^{LB}(s',a') - Q_{k+1}^{LB}(s',a') \right| \text{ (Lemma 3)}$$

$$\leq \gamma \| Q_k^{LB}(s,a) - Q_{k+1}^{LB}(s,a) \|_\infty$$

**Case 4:** If the second elements in max were the bigger values for both $Q_{k+1}$ and $Q_{k+2}$:

$$Q_{k+1}^{LB}(s,a) = \min_{s' \in \hat{T}(\cdot|s,a)} \left[ \mathcal{R}(s,a,s') + \gamma \max_{a'} Q_k^{LB}(s',a') \right]$$

$$Q_{k+2}^{LB}(s,a) = \min_{s' \in \hat{T}(\cdot|s,a)} \left[ \mathcal{R}(s,a,s') + \gamma \max_{a'} Q_{k+1}^{LB}(s',a') \right]$$

$$\left| Q_{k+1}^{LB}(s,a) - Q_{k+2}^{LB}(s,a) \right|$$

$$= \left| \min_{s' \in \hat{T}(\cdot|s,a)} \left[ \mathcal{R}(s,a,s') + \gamma \max_{a'} Q_k^{LB}(s',a') \right] - \max_{s' \in \hat{T}(\cdot|s,a)} \left[ \mathcal{R}(s,a,s') + \gamma \max_{a'} Q_{k+1}^{LB}(s',a') \right] \right|$$

$$\leq \gamma \| Q_k^{LB}(s,a) - Q_{k+1}^{LB}(s,a) \|_\infty \text{ (similar to Case 3)}$$

Since the above cases hold for any $s, a$, we therefore have:

$$\| Q_{k+1}^{LB} - Q_{k+2}^{LB} \|_\infty \leq \gamma \| Q_k^{LB} - Q_{k+1}^{LB} \|_\infty \tag{22}$$

Since the distance decreases by gamma with every iteration, it will converge to 0 and hence $Q^{LB}$ converges to a fixed point. $\qquad \square$

**Theorem 2** The Bellman operator in Q-M specifies only a non-strict contraction in general:

$$\left\| \mathcal{T}Q - \mathcal{T}\widehat{Q} \right\|_\infty \leq \left\| Q - \widehat{Q} \right\|_\infty$$

*Proof.* 1) For $\mathcal{T}_{\min}$ computing the upper bound:

$$\left| \mathcal{T}_{\min} Q(s,a) - \mathcal{T}_{\min}\widehat{Q}(s,a) \right| = \left| \min\left( Q(s,a), \max_{s'\in\widehat{T}(\cdot|s,a)} \left[ \mathcal{R}(s,a,s') + \gamma \max_{a'}(Q(s',a')) \right] \right) \right.$$

$$\left. - \min\left( \widehat{Q}(s,a), \max_{s'\in\widehat{T}(\cdot|s,a)} \left[ \mathcal{R}(s,a,s') + \gamma \max_{a'}(\widehat{Q}(s',a')) \right] \right) \right|$$

$$\leq$$

$$\max\left( \left| Q(s,a) - \widehat{Q}(s,a) \right|, \right.$$

$$\left| \max_{s'\in\widehat{T}(\cdot|s,a)} \left[ \mathcal{R}(s,a,s') + \gamma \max_{a'}(Q(s',a')) \right] \right.$$

$$\left. \left. - \max_{s'\in\widehat{T}(\cdot|s,a)} \left[ \mathcal{R}(s,a,s') + \gamma \max_{a'}(\widehat{Q}(s',a')) \right] \right| \right) \quad \text{(Lemma 4)}$$

$$\leq$$

$$\max\left( \left| Q(s,a) - \widehat{Q}(s,a) \right|, \right.$$

$$\left. \gamma \left| \max_{s'\in\widehat{T}(\cdot|s,a)} \max_{a'} \left[ Q(s',a') - \widehat{Q}(s',a') \right] \right| \right) \quad \text{(Lemma 3)}$$

$$\leq \max\left( \left\| Q - \widehat{Q} \right\|_\infty, \gamma \left\| Q - \widehat{Q} \right\|_\infty \right)$$

$$= \left\| Q - \widehat{Q} \right\|_\infty$$

2) For $\mathcal{T}_{\max}$ computing the lower bound:

$$
\left| \mathcal{T}_{\max} Q(s,a) - \mathcal{T}_{\max} \widehat{Q}(s,a) \right| = \left| \max\left( Q(s,a), \min_{s' \in \hat{T}(\cdot|s,a)} \left[ \mathcal{R}(s,a,s') + \gamma \max_{a'}(Q(s',a')) \right] \right) \right.
$$

$$
\left. - \max\left( \widehat{Q}(s,a), \min_{s' \in \hat{T}(\cdot|s,a)} \left[ \mathcal{R}(s,a,s') + \gamma \max_{a'}(\widehat{Q}(s',a')) \right] \right) \right|
$$

$$
\leq
$$

$$
\max\left( \left| Q(s,a) - \widehat{Q}(s,a) \right|, \right.
$$

$$
\left| \min_{s' \in \hat{T}(\cdot|s,a)} \left[ \mathcal{R}(s,a,s') + \gamma \max_{a'}(Q(s',a')) \right] \right.
$$

$$
\left. \left. - \min_{s' \in \hat{T}(\cdot|s,a)} \left[ \mathcal{R}(s,a,s') + \gamma \max_{a'}(\widehat{Q}(s',a')) \right] \right| \right) \quad \text{(Lemma 3)}
$$

$$
\leq
$$

$$
\max\left( \left| Q(s,a) - \widehat{Q}(s,a) \right|, \right.
$$

$$
\left. \gamma \left| \max_{s' \in \hat{T}(\cdot|s,a)} \max_{a'} \left[ Q(s',a') - \widehat{Q}(s',a') \right] \right| \right) \quad \text{(Lemma 4)}
$$

$$
\leq \max\left( \left\| Q - \widehat{Q} \right\|_\infty, \gamma \left\| Q - \widehat{Q} \right\|_\infty \right)
$$

$$
= \left\| Q - \widehat{Q} \right\|_\infty
$$

Since the above holds for any $s, a$ and for both $\mathcal{T}_{\min}$ and $\mathcal{T}_{\max}$, we have the conclusion holds. $\square$

**Theorem 3** [Optimality] For reward adaptation with Q variants, the optimal policies in the target domain remain invariant under Q-M when the upper and lower bounds are initialized correctly.

*Proof.* Let

$$
A_p(s) = \{\widehat{a} | \exists a \ Q^{LB}(s,a) > Q^{UB}(s,\widehat{a}); a \neq \widehat{a}\}
$$

$$
\tilde{A}(s) = A(s) \setminus A_p(s)
$$

where $A_p(s)$ represents the set of pruned actions under set $s$ and $\tilde{A}$ represents the remaining set of actions. To retain all optimal policies, it must be satisfied that none of the optimal actions under each state are pruned.

Assuming that a pruned action $\widehat{a}$ under $s$ is an optimal action, we must have

$$
\forall a \ Q^*(s,a) \leq Q^*(s,\widehat{a})
$$

Given that Q-M only prunes an action $\widehat{a}$ under $s$ when $\exists a \ Q^{LB}(s,a) > Q^{UB}(s,\widehat{a})$, we can derive that

$$
Q^{LB}(s,a) > Q^{UB}(s,\widehat{a}) \geq Q^*(s,\widehat{a}) \geq Q^*(s,a),
$$

resulting in a contradiction that

$$
Q^{LB}(s,a) > Q^*(s,a)
$$

As a result, we know that all optimal actions and hence policies are retained. $\square$

**Corollary** 1 [Non-uniqueness] The fixed point of the iteration process in Q-M may not be unique.

*Proof.* This can be proved using the following example:

Consider a three state MDP with states s1, s2, s3, where from s1 agent can take an action that transitions uniformly (0.5) to s2 and s3, from s2 agent can take an action that transitions uniformly (0.5) to s1 and s3, and s3 is the terminal state. Reward is 1 for both actions. There is no reward for the terminal state. Assuming a discount factor of 0.5.

For the upper bound, depending on how V(s3) is initialized, it may result in different fixed points:

- When V(s3) is initialized to a big value (say 4), a fixed point may be V(s1) = 3 and V(s2) = 3;

- When V(s3) is initialized to a small positive value (say 1), another fixed point could be V(s1) = 3/2 and V(s2) =3/2.

$\square$

## A.2 Algorithm

---
**Algorithm 1** Reward Adaptation via Q-Manipulation

---
1: Retrieve variants of $Q$'s, reachable states, and source reward functions from source domains.
2: Initialize $Q^{UB}$ and $Q^{LB}$ for the target behavior.
3: Tighten the bounds using the iteration process in Q-M.
4: Prune action.
5: Perform learning in the target domain with the remaining actions.

---

## A.3 Additional Information

### A.3.1 Domain Information

Detailed descriptions of the domains used for our evaluations are given below:
**Dollar-Euro:** A 45 states and 4 actions grid-world domain as illustrated in Fig. 1. **Source Domain 1 with $R_1$ (collecting dollars):** The agent obtains a reward of 1.0 for reaching the location labeled with "$", and 0.6 for reaching the location labeled with both $ and €. **Source Domain 2 with $R_2$ (collecting euros):** The agent obtains a reward of 1.0 for reaching the location labeled with €, and 0.6 for reaching the location labeled with both $ and €. **Target Domain with $\mathcal{R}$:** $\mathcal{R} = R_1 + R_2$.

**Frozen Lake:** A standard toy-text environment with 36 states and 4 actions. An episode terminates when the agent falls into any hole in the frozen lake (4 holes in total) or reaches the goal. **Source Domain 1 with $R_1$:** The agent is rewarded $+1$ for reaching any hole in a subset of holes (denoted by $H$), $-1$ for reaching any hole in the remaining holes (denoted by $\widehat{H}$) and 0.5 for reaching the goal. **Source Domain 2 with $R_2$:** The agent is rewarded $+1$ for reaching any hole in $\widehat{H}$, $-1$ for reaching any hole in $H$, and 0.5 for reaching the goal. **Target Domain with $\mathcal{R}$:** Avoid all the holes and reach the goal, or $\mathcal{R} = R_1 + R_2$.

**Race Track:** A 49 states and 7 actions grid-world domain. The 7 actions correspond to different velocities for going forward, turning left, or turning right. An initial location, a goal location, and obstacles make up the race track. An episode ends when the agent reaches the goal position, crashes, or exhausts the total number of steps. **Source Domain 1 with $R_1$ (avoid obstacles):** The agent obtains a negative reward of $-0.5$ for collision with a living reward of $+0.2$. **Source Domain 2 with $R_2$ (terminate):** The agent obtains a reward of $+2$ for reaching the goal, $-0.3$ living reward, and $-4$ for staying at the initial location. **Source Domain 3 with $R_3$ (stay put):** The agent obtains a reward of $+3$ for staying at the initial location. **Target Domain with $\mathcal{R}$:** Reach the goal in the least number of steps while avoiding all obstacles, or $\mathcal{R} = R_1 + R_2 + R_3$. This is the only domain where there are three source behaviors.

**Auto-generated Domains:** We have two different settings for auto-generating domains. These domains all feature two source domains and one target domain.

**Setting 1 (Designed Rewards)** Generate MDPs with the number of actions chosen randomly from $[9, 20]$ and the number of states chosen randomly from $[|A|, 80]$ where $|A|$ denotes the number of

actions. The transitions and transition distributions are then randomly generated. Initially, the number of reachable states from any $s, a$ is $|A|$. However, when an SBF is set for the generated MDP: for each $s, a$ pair, 1) we first randomly select a number $k$ from [1, SBF] as the number of reachable states from $s, a$, 2) we retain the state from the transition with the highest probability (which is often the "intended" state) while randomly choosing $k - 1$ states (without replacement) from its remaining reachable states; these are then considered as the new reachable states from $s, a$, and 3) re-normalize the transition distribution for $s, a$ based on these new reachable states. 3 states are randomly chosen to be the terminal states. Rewards for the source domains (i.e., $(R_1, R_2)$) for two of those states are set to $(+1, -1)$ and $(-1, +1)$, respectively; rewards for the third terminal state are set to $(+0.5, +0.5)$.

**Setting 2 (Randomized Rewards)** Here, we fix the number of states (50) and actions (8) for the generated MDPs. $R_1$ is sampled from a uniform distribution between [-5,-1) and $R_2$ is sampled from a uniform distribution between [1,5). Otherwise, we follow Setting 1.

For domains with continuous-state spaces, please note that the target domain may or may not follow the original environment's reward as described below:

**Ping-Pong:** We use a pygame pong environment. **Source Domain 1 (keep left)**: Agent is rewarded to keep left, negatively rewarded for keeping right, positively rewarded for scoring and penalized for opponent scoring. **Source Domain 2 (keep right)**: Agent is rewarded for keeping right, negatively rewarded for keeping left, positively rewarded for scoring and penalized for opponent scoring. **Target domain (win)**: The end goal is to keep scoring and prevent the opponent from scoring.

**Cartpole:** is a classic control gym environment. **Source Domain 1 ($\theta \leq -10$)**: Agent is rewarded for maintaining a large negative angle, penalized for a small negative angle, and mildly positively rewarded for living. **Source Domain 2 ($\theta \geq 10$)**: Agent is rewarded for maintaining a large positive angle, penalized for a small positive angle, and mildly positively rewarded for living. **Target domain**: Agent is rewarded for living (and thus maintaining the pole upright).

**Lunar Lander:** is a gym environment where we use: **Source Domain 1 (clockwise)**: agent is rewarded for tilting clockwise, penalized for tilting anti-clockwise, and positively rewarded for landing safely in the center. **Source Domain 2 (anti-clockwise)**: agent is rewarded for tilting anti-clockwise, penalized for tilting clockwise, and positively rewarded for landing safely in the center. **Target Domain**: the goal is to land safely in the center.

### A.4 Q-VARIANT

It is important to note that Q-variants may be difficult to learn with the same samples as experienced during a typical Q-learning process for $Q^*$. Some adaptation to Q learning must be made in order to learn $Q^*$ and $Q^\mu$ (or other Q-variant) via the same set of samples. Note that theoretically, Q learning is guaranteed to converge regardless of the behavior policy, although that is inefficient and can result in inaccuracy in practice due to that the behavior policy may result in visiting a different distribution of the states from that of the optimal policy (distributional shift). To ensure that $Q^*$ and $Q^\mu$ (or other Q-variant) can both receive informative samples, one possible way is to alternate between training $Q^*$ and $Q^\mu$ (or other Q-variant) and use importance sampling while using samples from $Q^\mu$ (or other Q-variant) to training $Q^*$ (or vice versa), so that we can leverage samples from both $Q^*$ and $Q^\mu$ (or other Q-variant) to train both $Q^*$ and $Q^\mu$ (or other Q-variant).

#### A.4.1 RUNNING TIME COMPARISON

We measured the running times taken to run each evaluation for each method for a fixed number of training steps on an XPS 9500 laptop. The aim here is to show that Q-M adds, in most cases, a reasonable amount of extra computation to the entire learning process. For Q-M, we considered the time from two main steps for each run (averaged over 30 runs): the iterative processes for tightening the bounds and Q-learning. For SFQL, we only considered the time taken for learning (the time needed for policy evaluation in SFQL was excluded). $|A_p|$ in the tables below, indicates actions pruned summed over all states.

| Domain | SBF | $|A_p|$ | | Time (s) | | | | | | |S| | |A| |
| | | mean | std | Q-M | | QL | | SFQL | | | |
| | | | | mean | std | mean | std | mean | std | | |
| Racetrack | 1 | 177.00 | 0.00 | 31.89 | 5.17 | **29.96** | 5.20 | 127.89 | 32.27 | | |
| | 5 | 58.20 | 12.85 | 36.88 | 6.57 | **30.13** | 5.25 | 104.89 | 22.85 | 49 | 7 |
| | 7 | 44.57 | 8.29 | 38.91 | 6.80 | **30.20** | 5.02 | 100.90 | 21.91 | | |
| Dollar-Euro | 1 | 104.00 | 0.00 | 5.79 | 0.79 | **4.77** | 0.70 | 5.56 | 1.49 | | |
| | 2 | 56.10 | 9.15 | 6.44 | 1.47 | **4.91** | 0.69 | 5.76 | 1.28 | 45 | 4 |
| | 4 | 30.20 | 5.90 | 6.68 | 0.92 | **5.35** | 0.84 | 5.97 | 0.94 | | |

Table 1: Running times for given MDP$\backslash R$ and designed rewards

| Domain | SBF | $|A_p|$ | | Time(s) | | | | | | |S| | | |A| | |
| | | mean | std | Q-M | | QL | | SFQL | | mean | std | mean | std |
| | | | | mean | std | mean | std | mean | std | | | | |
| Autogen Setting 1 | 1 | 499.63 | 254.40 | **15.03** | 5.86 | 15.56 | 8.23 | 53.94 | 51.11 | | | | |
| | 5 | 67.60 | 59.03 | 24.55 | 12.72 | **18.38** | 7.29 | 58.26 | 63.64 | 48.63 | 18.87 | 14.07 | 3.26 |
| | 9 | 26.40 | 44.46 | 27.56 | 11.88 | **21.59** | 9.20 | 51.29 | 42.70 | | | | |
| Frozen Lake | 1 | 76.47 | 1.61 | 36.66 | 9.58 | **28.45** | 10.56 | 40.89 | 11.33 | | | | |
| | 2 | 26.57 | 12.86 | 37.21 | 9.91 | **29.15** | 9.83 | 40.57 | 8.87 | 36.00 | 0.00 | 4.00 | 0.00 |
| | 4 | 7.37 | 4.00 | 37.31 | 10.36 | **36.26** | 9.69 | 40.36 | 14.31 | | | | |

Table 2: Running times for randomized MDP$\backslash R$ and designed rewards

| Domain | SBF | $|A_p|$ | | Time(s) | | | | | | |S| | |A| |
| | | mean | std | Q-M | | QL | | SFQL | | | |
| | | | | mean | std | mean | std | mean | std | | |
| Autogen Setting 2 | 1 | 329.24 | 1.30 | 1896.84 | 911.71 | 1722.13 | 435.75 | **540.63** | 248.75 | | |
| | 3 | 34.00 | 24.71 | 1469.28 | 798.21 | 1935.40 | 471.95 | **452.08** | 152.73 | 50 | 8 |
| | 5 | 10.52 | 8.15 | 1271.90 | 910.73 | 2199.68 | 49.31 | **362.75** | 31.59 | | |

Table 3: Running times for randomized MDP$\backslash R$ and randomized rewards

| Domain | SBF | $|A_p|$ | | Time(s) | | | | | | |S| | | |A| | |
| | | mean | std | Q-M | | QL | | SFQL | | mean | std | mean | std |
| | | | | mean | std | mean | std | mean | std | | | | |
| Autogen Setting 1 | 1 | 484.27 | 313.29 | **30.18** | 41.64 | 46.15 | 67.43 | 71.00 | 113.22 | | | | |
| | 3 | 72.17 | 120.38 | **43.42** | 37.20 | 52.45 | 58.70 | 63.41 | 89.53 | 51.43 | 20.45 | 14.37 | 3.30 |
| | 5 | 15.50 | 33.25 | **50.39** | 36.18 | 61.54 | 59.18 | 59.84 | 78.78 | | | | |

Table 4: Running times for nonlinear combination function

| Domain | SBF | $|A_p|$ | | Time(s) | | | | | | |S| | |A| |
| | | mean | std | Q-M | | QL | | SFQL | | | |
| | | | | mean | std | mean | std | mean | std | | |
| Autogen Setting 2 | 0 | 87.33 | 74.92 | 721.19 | 111.40 | **499.83** | 38.96 | 537.00 | 137.03 | | |
| | 0.25 | 32.03 | 16.18 | 689.61 | 110.06 | 528.35 | 13.70 | **484.81** | 138.65 | 50 | 8 |
| | 0.5 | 20.97 | 5.40 | 684.95 | 95.91 | 490.23 | 0.35 | **417.63** | 50.96 | | |

Table 5: Running times for noisy combination function

| Domain | SBF_min | SBF_mid | SBF_max |
| --- | --- | --- | --- |
| Dollar Euro | 0.05 | 0.04 | 0.03 |
| Race Track | 0.13 | 0.12 | 0.15 |
| Frozen Lake | 0.04 | 0.04 | 0.04 |
| Autogenerated | 0.11 | 0.20 | 0.18 |
| Autogenerated (randomized MDP and R) | 2.54 | 3.17 | 2.92 |
| Non-linear Target Reward | 0.95 | 3.83 | 5.25 |
| Noisy Reward Combination | 2.92 | 2.91 | 2.85 |

Table 6: Running time for Q-M iteration process

### A.4.2 ACTION PRUNING

For simulation domains, to understand the states where actions are pruned, we plot heat-maps. In all three simulation domains, we observe significant pruning around the terminal states. In addition, we also observe that fewer actions are pruned as SBF increases. The following color codes are used: initial state = yellow, goal states = green, terminal states/obstacles = black. We use different shades of blue to illustrate how many actions are pruned in a state: the lighter the color, the fewer the actions remained.

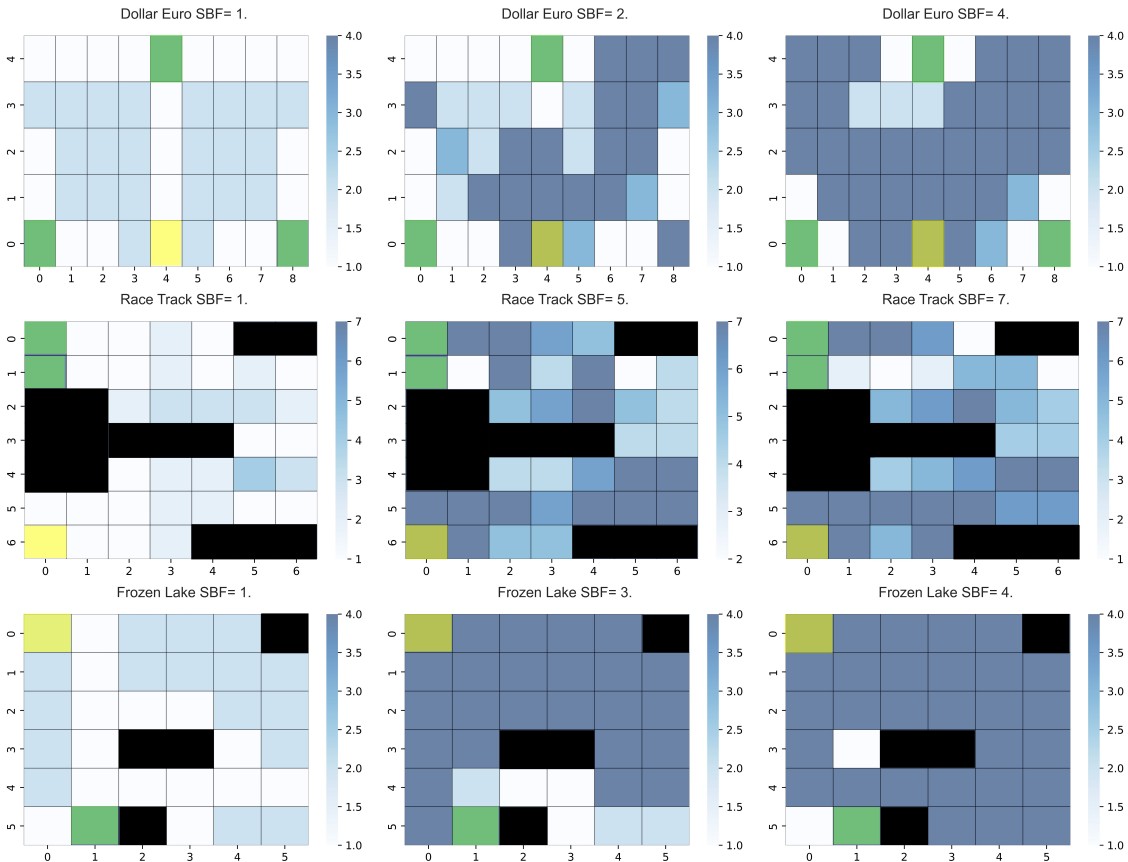

Figure 8: Heat-maps illustrating action pruning for a single run in simulation domains.

### A.4.3 HYPERPARAMETERS

All hyperparamters are set to be same for the different methods in the same evaluation domain. For continuous domains, the input layer of DQN is followed by 3 fully connected layers each consisting of $64$ neurons with relu activation. We used a buffer size of $100000$, a batch size of $64$, $\tau = 0.001$ for soft update of the target network parameters and a learning rate of $0.0005$. The exploration rate starts from $1.0$ and is gradually decayed in both discrete and continuous domains. $\gamma$ is chosen between $[0.9, 0.99]$ across different domains.

