# OpenReview forum: "Reward Adaptation Via Q-Manipulation"
_ICLR.cc/2025/Conference — Submitted to ICLR 2025_

### Official Review · Reviewer_GvVB · 2024-11-01

**Soundness:** 2
**Presentation:** 2
**Contribution:** 2
**Rating:** 5
**Confidence:** 3

**Summary:**

This paper considers the reward adaptation in reinforcement learning. The task is to learn optimal policy for target reward function given behavior data under source reward. The method is developed based on the assumption that the target reward is a known function of the source reward to derive the upper and lower bound of the Q function to perform action pruning. The proposed method is compared with successor feature based reward adaptation method in several domains.

**Strengths:**

+ Theoretical analysis on the Q bound

**Weaknesses:**

+ Problem Definition: The problem the authors aim to address is not clearly described. In Definition 1, reward adaptation is defined as the task of learning an optimal policy for a target reward function, given a set of behaviors trained under source reward functions. However, it is unclear whether the agent still has access to the underlying MDP when learning the target policy. Is the proposed method an offline algorithm where the target policy is learned only based on the source data?

+ Assumption. The assumption that the target reward function is a known function of the source rewards (e.g., a linear combination) is not well-motivated and seems overly restrictive. Additionally, the experimental domains are not well-described, making it difficult for readers to interpret this assumption. For instance, in Race Track, $R_3$ assigns a positive reward (+3) for remaining at the initial location.. This is a relatively large reward compared to others in $R_1$ and $R_2$. In the target domain when $\mathcal{R}=R_1+R_2+R_3$, factors like the distance between the goal and initial location, maximum episode length, and discount factor significantly influence policy behavior. In extreme cases, such as with a short maximum episode length and low discount factor, the optimal policy could be to remain at the initial location.

+ Plots. For the Q-M method, pre-training and pre-computation of Q-functions under the source reward are required. However, in the visualizations, it appears that the computational costs associated with this stage are not included in the plots. This omission raises concerns about the fairness of the comparisons.

**Questions:**

When the target reward is a linear combination of the source rewards, this setup appears quite similar to multi-objective reinforcement learning (MORL), where each objective is summed linearly. Could you clarify if there is any relationship between your method and multi-objective RL?

---

> ### Author Response · Authors · 2024-11-14
>
> Thank you for your valuable feedback. Please see our responses to your questions below:
>
> ### Clarification of Problem Statement
> - The agent doesn’t have access to the underlying MDP when learning the target policy but can interact with the target domain to collect samples. Q-M doesn’t have access to the source data. Source Q-functions (including Q variants for initializing the bounds) and that the target reward function is a known function of the source reward functions is the only information available.
>
> ### Motivation for $R = f(R_i)$
> - In practice, we often have a rough idea about how the target reward is related to the source rewards. For example, for adapting an autonomous driving agent to drive fast and comfortable when it already knows how to operate either fast or comfortable, we may consider a weighted linear relationship depending on the relative priority of each (a common approach in multi-objective optimization). Such a functional relationship, when unknown,  may also be learned first through regression.
>
> ### Domain Description
> - Experiments are designed such that source behaviors are different from target behaviors. For example, in race track individual behaviors correspond to avoid obstacles, terminate, or stay put. The target reward is -0.1 for living reward, -0.2 for colliding into the wall, -1 for staying at the initial location, and +2 for reaching the goal. The goal of the agent in the target domain is to avoid collision and reach the goal faster. Despite the source behaviors being very different from the target, Q-M is able to leverage it and achieve sample efficiency.
>
> ### Plots
> - When applying RL in practice, we often have sufficient computational capability but access to real-world samples is limited. In the introduction, we stated that “In general, Q-M requires additional computing resources (i.e., CPU time and storage) to implement but its benefits outweigh the costs in practical applications, especially in situations where accessing the target domain for samples is expensive.” Improvement in sample efficiency can be observed in the plots we present in section 3. At the same time, we reported running time comparison in the appendix, which shows that the increase in computation time is manageable.
>     - Time and samples used for policy evaluation in SFQL have not been considered in our results for SFQL; we assume that policy evaluation is essentially free for SFQL (which is true with some bookkeeping from the source behaviors). This also makes it fair to compare SFQL and Q-M since Q-M also assumes bookkeeping from the source behaviors (e.g., Q functions).
>
> ### MORL vs. Q-M
> - You are right in the sense that MORL and Q-M share some similarity. When there exists a function that combines the multiple objectives in MORL, the problem setting becomes similar to ours. When that is not possible, MORL introduces a pareto set of policies. Q-M is focused on the transfer learning aspect where source behaviors when available are leveraged to benefit learning the target behavior. MORL, on the other hand, addresses a pure optimization problem.

---

### Official Review · Reviewer_sg8A · 2024-11-03

**Soundness:** 4
**Presentation:** 3
**Contribution:** 3
**Rating:** 6
**Confidence:** 3

**Summary:**

This work proposes Q-Manipulation, a method for reward adaptation in RL which manipulates a Q-function to allow agents to adapt to new reward functions using prior learned behaviours. Upper and lower bounds for Q* for the new task can be determined when the combination function (the transform between the existing reward functions and the new one), which can be iteratively tightened and are guaranteed to reach a fixed point, similar to value iteration.

**Strengths:**

Overall, the strongest feature of this work is the theoretical guarantees provided on the convergence to a fixed point using the iterative method. This shows that the method is mathematically sound and motivates the extensions to noisy combination functions and continuous action/state spaces.

Pruning the unnecessary actions has the effect that the efficiency of the algorithm decreases with the stochastic branching factor of the MDP, leading to an interesting analysis of the SBF's contribution to convergence speed. As expected, convergence is instant with an SBF of 1, which is already an improvement over SFQL.

Empirically, Q-M outpaces the baselines in the discrete setting, with the linear combination function results being the most convincing. The robustness of the method to noisy reward functions is also convincing as the knowledge of the target reward function may be considered too restrictive in any practical settings.

The clarity of the work is good and the communications of the authors' claims and results are well-formulated.

**Weaknesses:**

The robustness of the method depends on the accurate initialisation of the Q bounds, and improperly placed bounds (which may occur if the target reward function is not well understood, for e.g.) might lead to inefficiency in the iterative pruning process. An analysis of the quality of the Q bounds to the final convergence speed in practice would be interesting.

Unfortunately, the good results in discrete environments don't carry over well to the continuous domain. The authors admit that the results for QM-DQN are only 'marginally better' for 2 environments, but even this is debatable for Lunar Lander. This is a crucial question for the future of this work because problem settings with known reward functions and convenient state discretisations may be difficult to find. In addition to the degradation of performance as the SBF increases, the scope of environments where Q-M is a viable solution seems limited. Have the authors considered other discretisation schemes, or perhaps a way to use Q-M in continuous environments without discretisation?

Finally, the lack of baselines makes the method a little hard to evaluate in the wider context of reward adaptation. However, as I was unable to identify any appropriate baselines apart from those in the paper, I cannot hold this against the authors as RA is largely an unexplored field as of yet.

**Questions:**

In Section 3.4, the following statement,

"In Pong, SF-DQN outperformed both QM-DQN and DQN. This was due to the choice of
source behaviors that are either keeping left or right. The target behavior requires the agent to move
to the left and right to catch the ball, which shares strong similarity with the source behaviors."

is confusing. Why doesn't the similarity between source and target functions also help the Q-M agent? Shouldn't the effects be felt more with QM-DQN than SF-DQN? If there is some insight here, a discussion on why QM-DQN didn't perform well would be useful to future readers as a gauge for determining which kinds of problems would be solved with either method.

---

> ### Author Response · Authors · 2024-11-14
>
> Thank you for your valuable feedback. Please see our responses to your questions below:
>
> ### Quality of the Q Bounds
> - It would be interesting to analyze the theoretical relation between the quality of bounds and sample efficiency in the future. In practice, we observed that the quality of bounds affected pruning performance, and hence sample efficiency. As evidence provided in our evaluation (section 3, pages 7-8), we showed that a noisy reward function, which resulted in looser bounds led to pruning fewer actions. However, whether looser bounds always lead to a reduced pruning performance requires a more thorough investigation.
>
> ### Continuous Domains
> - Extending Q-M to handle continuous domains (including studying other discretization methods) will be a future challenge so we provided only a simple discretization method to study its impact on the performance of Q-M. As we stated in the paper, discretization has a similar effect as adding noise (potentially unbounded) and hence the performance was expected to be substantially impacted. This observation is motivating us to seek more efficient methods for continuous domains. Currently, we are investigating how to apply Q-M with function approximation and working on a few directions.
>
> ### SFQL vs. Q-M with Source Behaviors Similar to the Target
> - For SFQL, if one of the source behaviors is similar to the optimal target behavior,  SFQL would be directly benefiting from it by starting from an almost optimal Q function. However, Q-M uses a very different mechanism for transferring knowledge: it only uses bounds of the Q functions to prune. The similarity between the source and target behaviors does not necessarily translate to tighter bounds. In fact, it could be quite the contrary (e.g., consider the linear case where the upper bound is a summation of all the source Q functions, Eq (9)). It is this very distinct mechanism that allows Q-M to handle cases efficiently (where the source and target behaviors are very different) while SFQL struggled. From this aspect, SFQL and Q-M are complementary to each other and can in fact be applied at the same time to combine their benefits for learning.

---

> > ### Comment · Reviewer_sg8A · 2024-11-21
> >
> > Thank you for your responses.
> >
> > As it stands, this work presents an interesting theoretical contribution that unfortunately does not reach that standard with its results. As the concerns of the Q-bounds and continuous domains are left for future investigation, I believe my rating should stay at its current value. The suggestions that "SFQL and Q-M are complementary to each other and can in fact be applied at the same time to combine their benefits for learning" would certainly be an interesting addition to this paper and I think it would be valuable to investigate how they interact.

---

> > > ### Author Response · Authors · 2024-11-21
> > >
> > > We thank you for your comments and suggestions on further studying the Q bounds, continuous domains, and the interaction between Q-M and SQFL, which all align with our future plan to extend the current theoretical work further.

---

> ### Author Response · Authors · 2024-11-21
> **QM+SFQL**
>
> We ran a quick evaluation with Q-M and SFQL on a similar setting with the Dollar Euro domain and observed results shown in the following link:
> https://ibb.co/99X4v4k
>
> This new result shows that running Q-M in combination with SFQL can indeed improve learning beyond what Q-M and SFQL individually can achieve
>
> we will incorporate this result in the paper and further study its effects in future work.

---

### Official Review · Reviewer_nCgP · 2024-11-04

**Soundness:** 2
**Presentation:** 3
**Contribution:** 2
**Rating:** 3
**Confidence:** 5

**Summary:**

In the context of tabular RL, the paper addresses the problem of "reward adaptation" - we have multiple policies trained on different source tasks (together with the corresponding optimal Q-values and the Q-values of the worst possible policy for each reward) and we want to produce an optimal policy for a target task whose reward is a known function of the source rewards.
The paper proposes a new method for this task called "Q-manipulation", which consists of establishing and then improving a lower and upper bound on the optimal Q-value for the target task. Where the lower bound for some actions is higher than the upper bound for other actions, this then allows us to prune the action space in each state, so when subsequently solving for the optimal Q-value on the target task, we have an easier task to solve.
The paper gives a Bellman iteration algorithm for refining the upper and lower bounds and shows that it converges to a (non-unique) fixed point. They also evaluate on several simple simulated tasks against Q learning without any reward adaptation and one prior method with reward adaptation.

**Strengths:**

- the writing is fairly clear with only minor typos
- the method seems to be a valid method for solving the task as defined (at least in the case where target reward is a linear function of the source rewards)
- the paper provides a valid theoretical result showing that the upper/lower bounds converge to a fixed point.

**Weaknesses:**

- The importance of task is not addressed so the paper lacks good motivation. No example of a practical real-world task where this could be useful is given.
- This becomes even more pressing given the assumptions of the proposed method. Firstly, even before the method is applied, it assumes we have access to the Q-values of the worst possible policy for each source task (in addition to the optimal ones). However, such worst-case Q-values are not something we usually naturally have in hand, and in general, obtaining each such worst-case Q-function means solving the whole RL problem as many times as there are source tasks. Instead, we could just solve for the target task and be done.
	- one could argue that this would be worth it in cases where we have a fixed set of source tasks, and we need to repeatedly adapt to different combinations of those, but this is not even mentioned.
- Ok, assume that for some reason, we have the worst-case Q-functions. Then, the proposed method still involves running a form of value iteration to refine the upper and lower bound on the target optimal Q-values. The paper doesn't show that this is cheaper than solving for the target task directly. Furthermore, for the method to make sense, it needs to be cheaper to first solve for this and then also run the subsequent Q-learning.
- The timing results hidden in the appendix show that on most tasks, this is indeed slower than running Q-learning from scratch, defeating the purpose of the method.
- Furthermore, I see hiding these important negative results in the appendix without mentioning them in the main text dishonest. The main article shows training curves with iterations that, for the proposed method, include only iteration spent after the expensive pre-training, which may give a wrong impression of computational savings.
- A fully valid end-to-end method is provided only for target rewards that are linear combinations of the source rewards - a fact that is again somewhat hidden in the text. I would like to see that emphasized earlier as it is an important limitation on the contributions of the paper.
- Corollary 1 is never proven. To prove it, you can just give an example of an MDP with non-unique fixed points, which could also serve as a useful illustration.
- Not enough details are provided for reproducing the experiments. Code is not provided.

**Questions:**

Questions:
- What are examples of real-world tasks on which your method could be practically useful? (if not in its current form, then at least aspirationally through future extensions)
- What is the break down of the training time using Q-M between the bound refinement and then the subsequent Q-learning?
- Do you disagree with any of the above outlined weaknesses and if so how?
- Why is the method called Q-manipulation? This doesn't seem particularly descriptive of what the method is doing (bounding the Q-function and pruning the actions) - in that light at least half of methods in RL (-adjacent) literature could be called Q-manipulation.


Minor comments and suggestions:
- The notion of "reachable states" is not properly defined/explained upon first usage. I'd recommend explaining as "reachable states" sometimes also refers to states that are eventually reachable (possibly over multiple steps).
- Before, or at least immediately after equations 5, 7, you should comment on the fact that we don't know Q*, otherwise it's really confusing.
- Table 5 in the appendix: I assume |A_p| is the total number of actions left summed over all states. Maybe it would be worth showing this as a percentage of the original number of actions?

---

> ### Author Response · Authors · 2024-11-14
>
> Thank you for your valuable feedback. Below, we address the concerns raised in the weaknesses and provide answers to the remaining questions:
>
> ### Motivation
> - Motivation is a new method of transfer learning for reward adaptation. A real-world example has been provided in the introduction: “adapting an autonomous driving agent to drive fast and safe when it already knows how to operate either fast or safe”. Our approach aims to reduce the samples required for learning the target behavior. It is achieved by pruning out actions that do not contribute to the optimal policy before learning.  Hence, Q-M can be used in an RL system that resembles modular RL where a repository of source behaviors (with additional bookkeeping for Q variants) are maintained and can be reused for learning new behaviors.
> ### Q Worst-Case
> - It is only needed for initialization. Our iteration method is valid if the bounds are initialized correctly. Other ways that do not require Q worse-case may exist and require future investigation. Eg.  to compute the bounds via the geometric sequence of the highest and lowest rewards (although it is expected to reduce the transferring performance). Assuming Q worse-case for the source behaviors is not necessarily strong assumption. In practice, Q worse-case can be obtained while learning the source behavior (with the same samples used for learning Q*) by leveraging Lemma 1. Storing it requires extra space but is feasible with modern computing. Note that we assume that these worst-case Qs are provided along with the source behaviors (so we do not need to learn them when learning the target behavior). This is not unlike modular RL where a repository of source behaviors can be utilized to form new behaviors but requires additional bookkeeping for the source behaviors (such as their initialization conditions) to apply.
> ### Running Time
> - Cost depends on the metric used. While you are right that our method is computationally expensive, it is less costly in terms of sample requirement.  Practically in RL, we often have sufficient computational capability but access to real-world samples is limited. In the introduction, we stated that “In general, Q-M requires additional computing resources (i.e., CPU time and storage) to implement but its benefits outweigh the costs in practical applications, especially in situations where accessing the target domain for samples is expensive.” Improvement in sample efficiency can be observed in section 3.
>     - We did not “hide” the computation time results. Line 249 in section 3 highlighted that we reported running time comparison in the appendix, which shows that the increase in computation time is manageable so we did not separately report in the paper (more details below).
> ### Linear Combination of Rewards
> - Linear Combination of Rewards is not a necessary condition for using Q-M. As long as the bounds are initialized correctly, the Q-M iteration remains sound. Linear combination is just a combination function for which we have provided an initialization method. Other efficient initialization methods for other forms of the combination function will be investigated in our future work.
> ### Corollary 1
> - Corollary 1 is a direct result from Theorem 1 (convergence to a fixed point) and Theorem 2 (non-strict contraction) so we did not provide a proof. Note that the fixed point may or may not be unique.  We have observed different fixed points in evaluation with different initializations as well.
> ### Training Time Breakdown
> - Time taken for target learning via QL in Q-M is similar to that taken by the baseline QL since they are run for the same number of steps to compare.
> - Time taken by the iteration process varies across domains (depending on no. of states, actions, SBF, initialization of bounds, etc). The time taken (in sec) by the iteration process was included in the computation time of Q-M in the tables in the appendix, which is also separately reported below for your convenience. Since the increase is not substantial, we did not separately report it in the paper.
> | Domain                        | SBF_min | SBF_mid | SBF_max |
> |-------------------------------|---------|---------|---------|
> | Dollar Euro                   | 0.05    | 0.04    | 0.03    |
> | Race Track                    | 0.13    | 0.12    | 0.15    |
> | Frozen Lake                   | 0.04    | 0.04    | 0.04    |
> | Autogenerated                 | 0.11    | 0.20    | 0.18    |
> | Autogenerated (randomized MDP and R) | 2.54 | 3.17 | 2.92 |
> | Non-linear Target Reward      | 0.95    | 3.83    | 5.25    |
> | Noisy Reward Combination      | 2.92    | 2.91    | 2.85    |
> ### Title
> - We perform our computation based on the source Q functions that are assumed to be given to benefit learning the target behavior.  It is a form of knowledge manipulating (instead of directly using) to enable transfer, in our case the Q functions, and hence named _Q manipulation_.
> ### Code
> - https://tinyurl.com/56pc3avf

---

> > ### Comment · Reviewer_nCgP · 2024-11-23
> >
> > Thank you for you response. Here are my reactions:
> >
> > **Motivation**: I don't find this motivating example plausible. Autonomous vehicles are generally trained directly to be both safe and (within the constraints of safety) appropriately fast. Your alternative approach would suggest (1) training an autonomous driving policy optimizing for safety only (this would plausibly be a policy that would just keep the car parked, or park it as soon as possible), (2) training a policy to be fast, but possibly unsafe (which would be, well, unsafe, though ok, assume we can do that in simulation), (3) training a policy to be as slow as possible (worst case policy for the speed-related reward function), (4) training another policy to be as dangerous as possible, (5) running a costly training procedure to deduce the bounds, (6) finally training a somewhat easier version of the safe&fast task. Beside, you don't provide any consideration of the roadmap between your initial small-scale steps and autonomous driving, so that we could consider whether it's a meaningful step in that direction.
> >
> > **Q Worst-Case**:
> > >  Q worse-case can be obtained while learning the source behavior (with the same samples used for learning Q*) by leveraging Lemma 1
> >
> > This is not true. The optimal policy for -R is generally a completely different policy than that for R, which may need to explore completely different parts of the state-action space. So no, same samples cannot be used for training both. I don't see how Lemma 1 helps. Finding such a worst-case policy requires a full separate run of an RL algorithm with similar expected cost to running RL on the primary reward R and again similar expected cost to running RL on the target task directly. And it is not a kind of policy and Q-function one naturally has at hand to start with, so I don't think simply assuming it's provided is fair, since it's an object specific to your method.
> >
> >
> > **Running Time**
> > Ok, so as I understand it, you concede the point on computational cost. However, you argue that your method is better in terms of "sample complexity". You say that "Practically in RL, [...] access to real-world samples is limited." Often, for a real world complex task, the scarcest resource for RL is human annotation time, i.e. reward signals. However, your Bellman operator for UB, LB assumes full access to the reward, so no savings there. Assuming access to reward is cheap (i.e. your task falls within the narrow range of real-world tasks where we have good automated evaluation), then the expensive resource are environment transitions. Here, you go half-way and assume partial knowledge of the environment dynamics: knowledge of reachable states, but not their exact probabilities (this again narrows the domain for your method). That setting seems somewhat narrow (again, I would appreciate practical justification when this is the case in a real-world setting), but accept that there exists a setup where we can consider your method to provide computational savings.
> >
> > **Linear Combination of Rewards**: Since valid initialization is needed, the only setting for which you provide a complete, ready-to-use method is the linear case. I do trust that future work could add initialization methods for other combination function, but his paper, as it stands, does not.
> >
> > **Corollary 1**: Just from a methodological perspective, the way you *prove* a non-uniqueness property is that you show that there may exist at least two different fixed points in some case. No such proof is given. No, it is not a direct consequence of the Theorems, especially since the "only non-strict" part of Theorem 2 is also never proven. If you prove that $a\leq b$, this does *not* imply that it is ever the case that $a=b$! You need to show that separately. The easiest way of doing that is providing an example.
> >
> > **Training Time Breakdown**: Thank you, the breakdown is useful to see.
> >
> > **Code**: Thank you for providing the code, this is helpful.

---

> > > ### Author Response · Authors · 2024-11-24
> > >
> > > Thank you for your additional comments, which are helpful. We respond to them below:
> > >
> > > ### Motivation:
> > > * Let us consider a slightly modified motivating example where a vehicle already knows how to drive 1) comfortably (in normal conditions with human passengers) and 2) drive fast (while only transporting goods where delivery time is to be minimized). In a new situation where the vehicle is transporting goods with human passengers onboard, the behavior must be both comfortable and fast. While we understand that there still are many gaps for running our system on real-world autonomous vehicles, that is NOT our goal, neither is autonomous driving our application domain. Our main goal here is to present a general and theoretical framework for reward adaptation (RA), under our assumptions. While the steps you listed are (partially, see below) correct, steps 1-4 are assumed to be coming from the source behaviors and not required in the target learning process.
> > >
> > >
> > > ### Q-worst case:
> > > * Sorry for the confusion here but we did not mean that you could learn $Q^*$ and $Q^{\mu}$ with the same samples as you would experience during a typical Q learning process for $Q^*$ . Some adaptation to Q learning must be made in order to learn $Q^*$
> > >  and $Q^{\mu}$ via the same set of samples. Note that theoretically, Q learning is guaranteed to converge regardless of the behavior policy, although that is inefficient and can result in inaccuracy in practice due to that the behavior policy may result in visiting a different distribution of the states from that of the optimal policy (distributional shift). To ensure that $Q^*$  and $Q^{\mu}$ can both receive informative samples, one possible way is to alternate between training $Q^*$  and $Q^{\mu}$ and use importance sampling while using samples from $Q^{\mu}$ to training $Q^*$ (or vice versa), so that we can leverage samples from both $Q^*$  and $Q^{\mu}$  to train both  $Q^*$  and $Q^{\mu}$. This works well in practice (one example using the Race Track domain is given here https://ibb.co/Hz4GfN3) without needing much more samples than the typical Q learning. Also, in practice, we often found that $Q^{\mu}$ is much easier to learn (e.g., learning to crash in lunar lander)
> > >
> > >
> > > ### Assumptions:
> > > * In the real-world, learning the dynamics is often much more difficult than learning the reward function, especially for systems with complex dynamics: under a given transition, while reward is a scalar, the domain dynamics is a distribution. When we can bound the uncertainty in the transition, we will essentially have access to the neighboring states (which is not unlike the motion uncertain model assumed in the Kalman filter and many others). We understand that these are still assumptions and our work may not be applicable in all possibly conceivable scenarios.
> > >
> > >
> > > ### Running time:
> > > * We would also like to point out that, in RL, cost is not just from humans providing the reward (such as in preference-based RL). Reward may be determined systematically from the environment (e.g., training a humanoid robot to move forward). More importantly in such situations, samples are needed to experience the dynamics of the environment, which would introduce wear and tear to the learning agent. Transition samples during learning the target behavior is what we aim to reduce via our approach.
> > >
> > >
> > > ### Corollary 1 example:
> > > * We can provide an example here. Consider a three state MDP with states s1, s2, s3, where from s1 we can take an action that transitions uniformly (0.5) to s2 and s3, from s2 we can take an action that transitions uniformly (0.5) to s1 and s3, and s3 is the terminal state. Reward is 1 for both actions. There is no reward for the terminal state. Assuming a discount factor of 0.5.
> > >
> > > * For the upper bound, depending on how V(s3) is initialized, it may result in different fixed points:
> > >
> > >     * When V(s3) is initialized to a big value (say 4), a fixed point may be V(s1) = 3 and V(s2) = 3;
> > >     * When V(s3) is initialized to a small positive value (say 1), another fixed point could be V(s1) = 3/2 and V(s2) =3/2.
> > >
> > >
> > > We will incorporate this example.

---

> > > > ### Comment · Reviewer_nCgP · 2024-12-03
> > > >
> > > > Thank you for your responses.
> > > >
> > > > **Motivation**: Now I see you partly retracting the only possible application direction you've given, so for me the work remains lacking good motivation. You claim 1-4 come from the source behaviour, but I've repeatedly raised the issue that 3-4 are not something one would naturally expect to have on a source domain. I haven't been given arguments to make me think otherwise.
> > > >
> > > > **Q-worst case**: Yes, since you're training an agent to optimize for literally the opposite goal, you can expect substantial distribution shift. Yes, in some domains, the worst-case policy may be easier to train and there may be some transfer of knowledge, but for me, it still remains another policy that needs to get trained.
> > > >
> > > > **Corollary 1 example**: Thank you. Yes, this finally proves your corollary.
> > > >
> > > > **Conclusion**: I'm keeping my score mainly due to the usefulness of this work still remaining dubious to me - the assumptions the work builds on seem strange, as expressed earlier, and the usefulness hasn't been well explained by the authors.

---

> > > > > ### Author Response · Authors · 2024-12-03
> > > > >
> > > > > Thank you for the additional comment.
> > > > >
> > > > > **Motivation**: To use our approach as a modular RL method, we assume that the source domain provides $Q^*$ and $Q^{\mu}$. While this is not consistent with the output of traditional Q-learning, we explained that an adaptive Q-learning can be used without much additional cost.
> > > > >
> > > > > **Q-worse case**: We discussed an importance-sampling based approach and provided evidence that the two policies could be learned simultaneously using roughly the same amount of samples as traditional Q-learning.

---

### Author Response · Authors · 2024-11-13

We would like to express our sincere gratitude to all the reviewers for their invaluable feedback on our work.

We would like to point out that, while there are limitations (due to some of the assumptions made), our approach to reward adaptation supports more general knowledge transfer and is fundamentally different from and complementary to the previous work. It can be used by itself to benefit learning the target behavior or as a key part of a larger system that resembles modular RL, and thus has many real-world applications (more details in individual responses). Our main contributions hence lies in theoretically and empirically establishing the feasibility and effectiveness of Q-M and opens up many future research opportunities. We would like to stress that the assumptions only incur manageable additional costs in practice (while allowing Q-M to substantially reduce the samples required for training the target behavior) and are not very different from similar assumptions made in relevant prior works (more details in individual responses).

Next, we will address individual questions and concerns in our responses to individual reviewers.

---

> ### Author Response · Authors · 2024-11-19
>
> Since it has been a week since we posted our responses, we would like to send a friendly reminder and check whether our responses addressed your concerns.

---

### Author Response · Authors · 2024-11-28

We have updated the paper and appendix to address the reviewers’ current set of comments (changes in blue) and would be happy to respond to any further concerns/questions.

---

### Meta-Review · Area_Chair_UwzX · 2024-12-20

**Metareview:**

This paper proposes a new method for reward augmentation that aims to adapt the behavior of the agent based on known prior behaviors. The paper provides theoretical support proving the convergence of the algorithm in tabular settings and some empirical demonstrations. The reviewers are not positive on the paper, citing a lack of motivation for which the problem setting is relevant, a setting for which the method is even practical, and empirical results that suggest that the method is not even providing sufficient benefit in the chosen environments. Altogether, it makes the contributions of this paper (a novel algorithm) unclear as to who or when someone would benefit from understanding it. The authors should take the reviewer's perspective into account when reformulating their paper.

**Additional Comments On Reviewer Discussion:**

There was sufficient back and forth between the authors and reviewers, but the reviewers' concerns were not addressed adequately. Based on these unaddressed concerns, I recommended the paper for rejection.

---

### Decision · Program_Chairs · 2025-01-22

Reject